# DAL: A Practical Prior-Free Black-Box Framework for Piecewise Stationary Bandits

**Argyrios Gerogiannis** [1]  **Yu-Han Huang** [1]  **Subhonmesh Bose** [1]  **Venugopal V. Veeravalli** [1]

## Abstract

We introduce a practical, black-box framework termed Detection Augmented Learning (DAL) for the problem of piecewise stationary bandits without knowledge of the underlying non-stationarity. DAL accepts any stationary bandit algorithm with order-optimal regret as input and augments it with a change detector, enabling applicability to all common bandit variants. Extensive experimentation demonstrates that DAL consistently surpasses all state-of-the-art methods across diverse non-stationary scenarios, including synthetic benchmarks and real-world datasets, underscoring its versatility and scalability. We provide theoretical insights into DAL's strong empirical performance, complemented by thorough empirical validation.

## 1. Introduction

Bandit models underpin a wide range of engineering systems, from recommendation and ads to dynamic pricing and real-time bidding (Li et al., 2010; Schwartz et al., 2017; Sertan et al., 2012; Tajik et al., 2024; Flajolet & Jaillet, 2017). Many variants of bandits have emerged since the work of Robbins (1952), which fall into parametric bandits (PBs) (Auer, 2002; Faury et al., 2020; Filippi et al., 2010), non-parametric bandits (Srinivas et al., 2010) and contextual bandits (CBs) (Woodroofe, 1979; Langford & Zhang, 2007). In the general bandit problem, in each round, an agent receives a context $C_t$ randomly sampled from a set $\mathcal{C}$, and selects a policy $\pi_t$ from a policy set $\Pi$—a set of mappings from $\mathcal{C}$ to a compact action set $\mathcal{A} \subseteq \mathbb{R}^d$. Then, the agent chooses action $A_t = \pi_t(C_t)$ and receives reward

$$X_t = f_t(C_t, A_t) + \varepsilon_t,$$

[1]Department of Electrical and Computer Engineering, The Grainger College of Engineering, University of Illinois at Urbana-Champaign, Champaign, USA. Correspondence to: Argyrios Gerogiannis <ag91@illinois.edu>.

*Proceedings of the 43rd International Conference on Machine Learning*, Seoul, South Korea. PMLR 306, 2026. Copyright 2026 by the author(s).

where $f_t : \mathcal{C} \times \mathcal{A} \to \mathbb{R}$ is the reward function and $\varepsilon_t$ is the zero-mean sub-Gaussian noise. The goal is to minimize the dynamic regret, using a causal policy $\pi_t$:

$$R_T := \mathbb{E}_{\substack{A_t \sim \pi_t \\ C_t \sim \mathcal{P}_t}} \left[ \sum_{t=1}^{T} \max_{\pi \in \Pi} f_t(C_t, \pi(C_t)) - f_t(C_t, A_t) \right].$$

CBs follow the general formulation above, where the context $C_t$ is independently sampled from $\mathcal{P}_t$ and $|\mathcal{A}|$ is finite. In parametric and non-parametric settings, the context is fixed across time and $|\mathcal{A}|$ can be infinite, and with slight abuse of notation, we write $f_t(C_t, A_t) = f_t(A_t)$. For parametric bandits, $f_t(A_t) = \mu(\langle \theta_t, A_t \rangle)$, where $\theta_t$ is a bounded unknown parameter and $\mu : \mathbb{R} \to \mathbb{R}$ is injective. These include linear bandits (LBs), with $\mu$ as identity, generalized linear bandits (GLBs), and self-concordant bandits (SCBs), where $\mu$ is self-concordant and the noise variance may depend on the mean (Russac et al., 2021). For non-parametric, we consider kernelized bandits (KBs), where $f_t \in H_k$, a reproducing kernel Hilbert space (RKHS) induced by a continuous positive semi-definite kernel $k : \mathcal{A} \times \mathcal{A} \to \mathbb{R}$ with $k(x, x) \leq 1$ and $\|f_t\|_{H_k} \leq B$. In KBs, a central complexity measure is the maximum information gain $\gamma_T$ (worst-case mutual information between $f$ and $T$ noisy evaluations). For compact $\mathcal{A} \subset \mathbb{R}^d$: $\gamma_T = \mathcal{O}((\log T)^{d+1})$ for the Squared Exponential (SE) kernel, and $\gamma_T = \mathcal{O}(T^\beta \log T)$ with $\beta = d(d+1)/[2\nu + d(d+1)]$ for Matérn($\nu$) kernels.

Bandits remain practically relevant today: recent deployments span A/B testing (Zhang et al., 2025), clinical trials (Varatharajah & Berry, 2022), large language models (Shin et al., 2025), diffusion models (Aouali, 2024), and computer architecture (Gerogiannis & Torrellas, 2023), which even leverage the canonical formulations as the core decision engine. Accordingly, the key challenge is developing bandit methods that perform reliably under real-world constraints—aimed at practical effectiveness, not just analysis. The lion's share of the literature on bandits assumes *stationarity*—i.e., fixed $f_t$, $\theta_t$, $\mathcal{P}_t$—but this rarely holds in practice due to evolving conditions (Agrawal & Jia, 2019; Cai et al., 2017; Chen et al., 2020; Lu et al., 2019). Non-stationary (NS) settings are often categorized into two types–*gradual drifts* and *abrupt changes*. In the drifting model, $f_t$ and $\mathcal{P}_t$ evolve slowly under a variation budget

constraint (Besbes et al., 2014; Wei & Luo, 2021). In contrast, piecewise stationary (PS) models assume abrupt shifts at unknown change-points, with a total of $N_T$ changes:

$$1 =: \nu_0 < \nu_1 < \cdots < \nu_{N_T} < \nu_{N_T+1} := T+1$$

with $f_t = f_{t'}$ and $\mathcal{P}_t = \mathcal{P}_{t'}$ for $t, t' \in \{\nu_k, \dots, \nu_{k+1} - 1\}$ and different across change-points.

NS bandit algorithms are typically either *adaptive*, adjusting continuously, or *restarting*, choosing to unlearn and kickstart the learning process at certain times. They may also be *prior-based* (assuming knowledge of the non-stationarity) or *prior-free*. Prior-based adaptive methods (discounting/sliding window) weigh recent observations more heavily and have been developed for multi-armed bandits (MABs) (Garivier & Moulines, 2011; Kocsis & Szepesvári, 2006), LBs (Cheung et al., 2019; Russac et al., 2019), GLBs (Faury et al., 2021; Russac et al., 2020), SCBs (Russac et al., 2021; Wang et al., 2023), KBs (Deng et al., 2022; Zhou & Shroff, 2021). Prior-based restarting approaches use budgeted restarts and have been studied for MABs (Besbes et al., 2014), LBs/GLBs (Zhao et al., 2020), KBs (Zhou & Shroff, 2021). Detection-based restarting methods exist in both flavors: prior-based for MABs (Cao et al., 2019b; Liu et al., 2018); prior-free for MABs (Auer et al., 2019; Besson et al., 2022; Huang et al., 2025), for LBs/KBs (Hong et al., 2023) and for CBs (Chen et al., 2019). The most closely related work is (Huang et al., 2025), which addresses PS-MABs and introduces techniques that we build upon in establishing our theory.

The majority of approaches in the literature are prior-based, and since non-stationarity cannot be known in advance, it is not possible to employ them in practice. The only prior-free alternatives detect non-stationarity by checking for violations of stationary regret bounds. MASTER (Wei & Luo, 2021) is the most prominent method of this class, as it is the only prior-free, order-optimal method for general bandits and reinforcement learning. Importantly, MASTER is a black-box method. Among prior-free methods, the black-box paradigm is particularly appealing: it equips *any* stationary bandit algorithm with non-stationarity handling capabilities. Although MASTER is order-optimal, it is not practically applicable, as its detection mechanism requires a horizon of at least 143 billion before it has any chance of triggering (Gerogiannis et al., 2025). More broadly, the literature emphasizes theory over evidence, as empirical validation of order-optimal methods is scarce: NS parametric and non-parametric bandits are evaluated almost exclusively on synthetic data (Wang et al., 2023; Hong et al., 2023; Gerogiannis et al., 2025), and contextual bandits lack experiments altogether (Chen et al., 2019). We close these gaps with a theoretically grounded, practical black-box framework and comprehensive real-world evaluation.

Our black-box turns any order-optimal stationary algorithm into an order-optimal piecewise stationary one by augmenting it with any detector satisfying a specific property, which enables flexibility in detector choice. Crucially, compared to all existing prior-free methods, our method is the first method for general bandits that detects actual changes in the reward model instead of violations of stationary regret bounds, something that also performs significantly better empirically. This "detect-the-model" paradigm is fundamentally different from all prior work and enables practical effectiveness.

**Contributions.** We present (to our knowledge) the first *practical* prior-free, black-box detection-based framework for general PS bandits. The design is motivated by three pragmatic insights: (i) prior knowledge of non-stationarity is rarely available, (ii) restart-style methods can have lower worst-case complexity than fully adaptive schemes (Peng & Papadimitriou, 2024), and (iii) a black-box reduction simplifies non-stationary algorithm design to specifying when to restart a stationary learner. Our method is simple—combining a change detector with any stationary bandit algorithm—modular, and easy to implement. Empirically, extensive synthetic and real-world evaluations on standard datasets show consistent gains over both prior-free and prior-based baselines, and (to our knowledge) provide the first comprehensive real-world assessment of order-optimal baselines previously lacking empirical study. Theoretically, under mild assumptions, our regret matches the state-of-the-art for PS-LBs, PS-GLBs and PS-CBs and *improves* the best known bounds for PS-SCBs and PS-KBs; for drifting regimes we identify conditions for good performance and validate them empirically.

## 2. The DAL Framework

The DAL framework is a black-box characterized by a modular structure of three components: a non-stationarity detector, a forced exploration scheme, and a bandit algorithm. We provide high-level ideas of the structure of our approach and formally present our framework in Algorithm 1.

**Non-Stationarity Detector** To identify changes in the environment, DAL uses a general-purpose detector $\mathcal{D}$ for monitoring shifts in the distribution of judiciously chosen reward observation sequences obtained through forced exploration. This distinguishes our approach from all existing methods, which rely on detecting violations of stationary regret guarantees. We adopt a detector aligned with (Besson et al., 2022; Huang et al., 2025), grounded in the well-established theory of quickest change detection (Veeravalli & Banerjee, 2014; Xie et al., 2021). Given any arbitrary context, DAL samples rewards from actions within a carefully selected finite subset, and detects changes in the mean reward associated with the context-action pair.

**Algorithm 1** **D**etection **A**ugmented **L**earning (**DAL**)

---

**Input**: bandit $\mathcal{B}$, detector $\mathcal{D}$, covering set $\mathcal{A}_e = \{a^{(i)} : i \in [N_e]\}$, context set $\mathcal{C}$, frequencies $\{\alpha_k\}_{k=1}^T$, horizon $T$
**Initialize**: histories $\mathcal{H}_{(c,a)} \leftarrow \emptyset \;\; \forall (c,a) \in \mathcal{C} \times \mathcal{A}_e$, detection step $\tau \leftarrow 0$, counter $k \leftarrow 1$, covering set size $N_e \leftarrow |\mathcal{A}_e|$

1: **for** $t = 1, 2, \ldots, T$ **do**
2:     Observe context $C_t$
3:     **if** $((t - \tau - 1) \mod \lceil N_e/\alpha_k \rceil) + 1 = i \in [N_e]$ **then**
4:         Play $i$-th action in $\mathcal{A}_e \rightarrow a^{(i)}$ and receive reward $X_t$
5:         Add reward $X_t$ to history $\mathcal{H}_{(C_t, a^{(i)})}$
6:         **if** $\mathcal{D}\left(\mathcal{H}_{(C_t, a^{(i)})}\right) =$ detection **then**
7:             Reset the bandit algorithm $\mathcal{B}$
8:             Clear all $\mathcal{H}_{(c,a)} \;\; \forall (c,a) \in \mathcal{C} \times \mathcal{A}_e$,
9:             $\tau \leftarrow t, \quad k \leftarrow k + 1$
10:    **else**
11:        Run the stationary bandit algorithm $\mathcal{B}$

---

**Forced Exploration** In stationary settings, effective algorithms quickly concentrate on optimal actions for each context, rarely exploring suboptimal actions. In NS environments, however, this behavior may lead to missed changes on these rarely sampled actions, and thus, *forced exploration* on these actions is essential. When the action space is large or infinite, exploring all actions becomes infeasible. Therefore, DAL only does extra exploration on a finite *covering set*, $\mathcal{A}_e = \{a^{(i)} : i \in [N_e]\} \subseteq \mathcal{A}$, in which $a^{(i)}$ denotes the $i$-th action in $\mathcal{A}_e$. $\mathcal{A}_e$ is designed such that the mean reward of at least one context-action pair in $\mathcal{C} \times \mathcal{A}_e$ changes whenever a change occurs. In particular, after the $(k-1)^{\text{th}}$ restart, DAL is forced to play each action in $\mathcal{A}_e$ once for $N_e$ steps, followed by the bandit algorithm for the next $\lceil N_e/\alpha_k \rceil - N_e$ steps, repeatedly, until the $k^{\text{th}}$ restart. Here, $\alpha_k \in (0, 1)$ is the exploration frequency, striking a balance between detection delay and regret from extra exploration.

**Bandit Algorithm** With a detector $\mathcal{D}$ and forced exploration, DAL augments a (stationary) bandit algorithm $\mathcal{B}$: it resets $\mathcal{B}$ entirely whenever $\mathcal{D}$ detects changes in a reward distribution associated with any context-action pair in $\mathcal{C} \times \mathcal{A}_e$, and runs $\mathcal{B}$ with periodic forced exploration otherwise. A key advantage of DAL is its ability to translate strong stationary performance into robust performance under NS conditions. Therefore, by selecting a well-performing bandit algorithm, the DAL framework inherently achieves effective adaptation to NS environments. The only requirement for DAL's input stationary algorithm is to attain optimal stationary regret performance bounds.

# 3. Practical Performance

## 3.1. Experimental Baselines

We evaluate *all* methods referenced in Section 1, highlighting the strongest state-of-the-art algorithms applicable to PS and drifting settings across both synthetic and real-world benchmarks. These baselines include MASTER (Wei & Luo, 2021), the only black-box method with order-optimal regret. MASTER lacks guarantees for NS-SCBs, but empirical evidence (Wang et al., 2023) supports using it with Log-UCB-1 (Faury et al., 2020). We additionally include two prior-free, order-optimal algorithms: ADA-OPKB (Hong et al., 2023) for NS-LBs/NS-KBs and ADA-ILTCB+ (Chen et al., 2019) for NS-CBs. ADA-OPKB requires extensive tuning (7 hyperparameters), which is incompatible with a fully prior-free setting; nevertheless, we tune it (and MASTER's single parameter $n$) for best performance. We also compare against two prior-based discounted approaches, WeightUCB (Wang et al., 2023) for drifting PBs and PS-SCBs, and WGP-UCB (Deng et al., 2022) for drifting KBs. All remaining methods are *prior-based*. To maintain readability, we group algorithms by paradigm (discounted, sliding-window, budget-restart) while keeping distinct methods separate when they differ meaningfully. In real-world experiments, we focus on the strongest current state-of-the-art methods, as the remaining algorithms are less competitive. We use the hyperparameters specified in the original works.

## 3.2. Practical Tuning of DAL

**Detector and Bandit Selection** Across all settings, DAL uses the *Generalized Likelihood Ratio (GLR)* and the *Generalized Shiryaev-Roberts (GSR)* tests (Huang & Veeravalli, 2025) as the detector $\mathcal{D}$, which are given in Algorithms 2 and 3. For the detectors, we set their thresholds $\beta_{\text{GLR}}(n, \delta_F) = \log(n^{3/2}/\delta_F)$ and $\beta_{\text{GSR}}(n, \delta_F) = n^{5/2}/\delta_F$, with $\delta_F = 1/\sqrt{T}$, as per Huang et al. (2025); Besson et al. (2022). Concretely: In LBs, LinUCB (Abbasi-yadkori et al., 2011) pairs with Gaussian GLR and GSR. In GLBs, GLM-UCB (Filippi et al., 2010) pairs with Gaussian GLR and GSR. In SCBs, OFUGLB (Lee et al., 2024) pairs with Bernoulli GLR and GSR. In KBs, REDS (Salgia et al., 2024) pairs with Gaussian GLR and GSR. In CBs, SquareCB (Foster & Rakhlin, 2020) pairs with Bernoulli GLR and GSR. We implement the stationary bandit algorithms as per their original works. For all settings, we set $\alpha_k = \sqrt{k|\mathcal{C}|N_e}/(2\sqrt{T}\log^2 T)$ as per Theorem 4.8. A crucial advantage of DAL is that it is *hyperparameter-free*, guided entirely by our theoretical principles.

**Construction of** $\mathcal{A}_e$ To construct $\mathcal{A}_e$, for PBs we follow Proposition 4.2: we greedily select linearly independent actions until collecting $d$, or as many as exist if fewer than $d$ are available. For KBs, $\mathcal{A}_e$ is from a $\delta_T$-cover by selecting the centers of the covering balls according to Proposition 4.3 and Corollary 4.9. In finite action spaces, we compute $\gamma_T$; if $|\mathcal{A}| \leq \gamma_T$, then by Corollary 4.9 we take $\mathcal{A}_e = \mathcal{A}$, otherwise we select the $\gamma_T$ actions closest to the cover centers. In all our kernelized experiments, $\gamma_T$ is larger than $|\mathcal{A}|$, so we always have $\mathcal{A}_e = \mathcal{A}$. In CBs, the action space is fi-

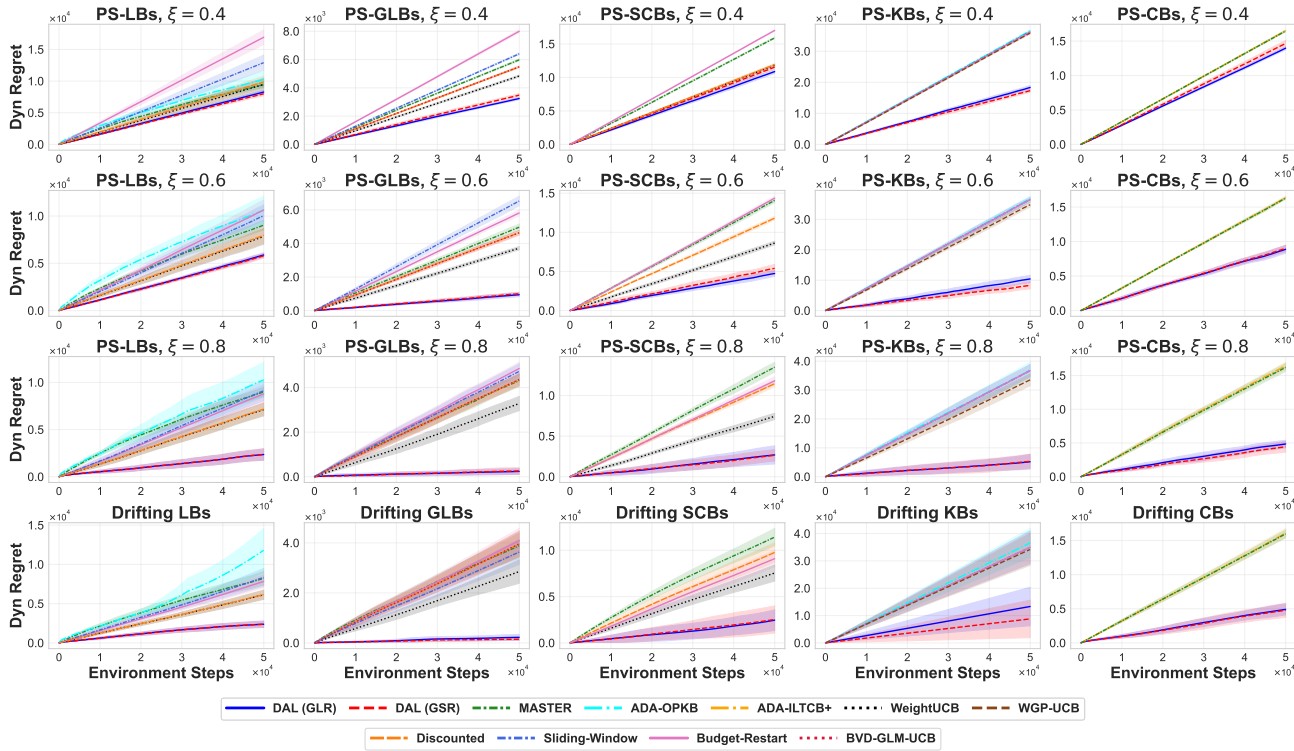

*Figure 1.* Dynamic regret vs. environment steps for synthetic experiments (lower=better). First three rows correspond to the geometric change-points and the final one to the drifting case. DAL attains the lowest dynamic regret across all different scenarios.

nite, and as noted in Remark 4.4, both the theory and our experiments take $\mathcal{A}_e = \mathcal{A}$. DAL's exploration burden is $N_e = |\mathcal{A}_e|$, which is determined by the structural complexity of the reward class. We emphasize that when $|\mathcal{A}_e|$ is large, the non-stationary bandit problem itself is already difficult, and thus, the cardinality of the covering set is not a bottleneck of DAL. In parametric and kernelized bandits $|\mathcal{A}_e|$ is independent of the infinite $\mathcal{A}$, while in CBs all *finite* actions must be explored. DAL limits $N_e$ to the minimum needed to characterize the reward function for detection and learning. DAL is driven by structural complexity, not $|\mathcal{A}|$. An extended discussion on $\mathcal{A}_e$ and its implications appears in the Appendix.

### 3.3. Synthetic Experiments

#### 3.3.1. EXPERIMENTAL PARAMETERS

**Common parameters** In all synthetic experiments, the action space comprises 100 unique actions with dimension $d = 10$. These actions are sampled independently from $\mathcal{N}(0, I)$. The horizon is fixed to $T = 50000$ and we average the results over 15 independent trials.

**Remark 3.1.** *In Algorithm 1, when $|\mathcal{A}|$ is finite, change-detection can be performed on the actions selected by $\mathcal{B}$ that are not in $\mathcal{A}_e$, which improves performance. This does not affect the theoretical properties of the algorithm, and*

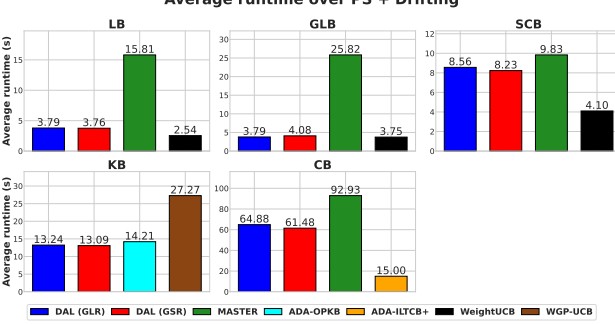

*Figure 2.* Average runtime in seconds for the top-performing algorithms, averaged over PS and drifting simulations.

*we employ this variation for our experiments.*

**PBs.** Actions lie in an $L$-ball and the parameters $\theta_t$ in an $S$-ball. We set $S = L = 1$ for LBs/GLBs and $(L, S) = (1, 3)$ for SCBs. At each initialization/change, entries of $\theta_t$ are drawn i.i.d. $\text{Unif}[-1, 1]$ and then rescaled to satisfy $\|\theta_t\| \leq S$. We use $\mu(x) = (1 + e^{-x})^{-1}$. Noise is $\varepsilon_t \sim \mathcal{N}(0, 0.01)$ per round, and the SCB rewards are sampled as $\text{Bernoulli}(\mu(\langle \theta_t, A_t \rangle))$. We set $\mathcal{A}_e$ via Corollary 4.9.

**KBs.** Actions lie in the $\sqrt{d}$-ball and $\varepsilon_t \sim \mathcal{N}(0, 0.01)$. We use the SE kernel with $\ell = 0.2$ and follow (Deng et al., 2022): discretize $[-1, 1]$ into 200 evenly spaced points

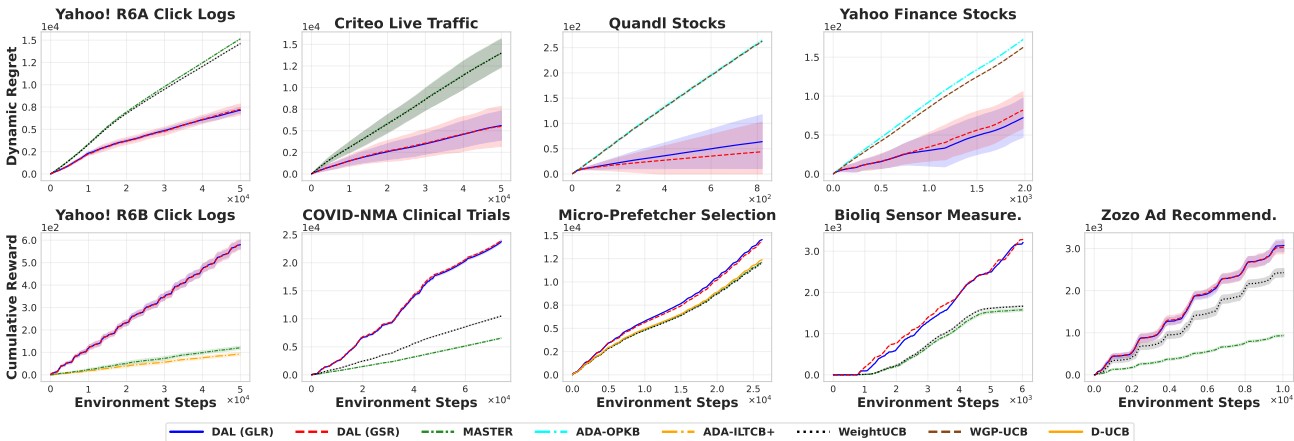

*Figure 3.* Results for real-world experiments of Section 3.4, averaged over 15 independent runs. Top: dynamic regret (lower=better); Bottom: cumulative reward (higher=better). DAL attains the best performance in all real-world datasets.

$\{x_i\}_{i=1}^{200}$ and generate $f_t$ from the corresponding RKHS via $f(\cdot) = \sum_{i=1}^{200} \alpha_i k(\cdot, x_i)$ with $\alpha_i \sim \mathrm{Unif}[-1,1]$. We set $\mathcal{A}_e$ via Corollary 4.9.

**CBs.** The contexts $C_t \in \mathbb{R}^{d_c}$ with $d_c = 10$ are randomly drawn each round from a fixed pool of 1000 normalized vectors. At every initialization/change, at least one of $f_t$ or $\mathcal{P}_t$ changes. For $a \in \mathcal{A}$, define the $[0,1]$−clipped $f_t(C_t, a)$,

$$\left[ b_a + z^{(\sigma)} \sigma(u_a^\top C_t) + z^{(s)} \sin(v_a^\top C_t) + z^{(x)} C_{t,2} C_{t,3} \right]_{[0,1]},$$

where $u_a, v_a \sim \mathcal{N}(0, I)$, $b_a \sim \mathrm{Unif}[0.3, 0.7]$, and $z^{(\sigma)}, z^{(s)}, z^{(x)}$ are drawn uniformly from $[0.25, 0.45]$, $[0.15, 0.35]$, $[0.10, 0.25]$, respectively. Rewards are sampled as $\mathrm{Bernoulli}(f_t(C_t, A_t))$. Since there is no arm-related structure, we set $\mathcal{A} = \mathcal{A}_e$ (Remark 4.4).

### 3.3.2. EXPERIMENTAL BENCHMARKS

**Piecewise Stationarity** We adopt the geometric change-point model (Gerogiannis et al., 2025), and sample the intervals between the change-points according to a geometric distribution with parameter $\rho = T^{-\xi}$, for $\xi \in \{0.4, 0.6, 0.8\}$. We do not impose any restriction on the lengths of the intervals between change-points in our experiments.

**Drifting Non-Stationarity** We drift linearly over $T$ rounds from an initial value to a final value, where the end-points are chosen as in the beginning of the section. Specifically,

$$\text{Parametric: } \theta_t = (1 - t/T)\theta_{\mathrm{init}} + (t/T)\theta_{\mathrm{final}},$$
$$\text{Kernelized: } f_t = (1 - t/T)f_{\mathrm{init}} + (t/T)f_{\mathrm{final}},$$
$$\text{Contextual: } \phi_t = (1 - t/T)\phi_{\mathrm{init}} + (t/T)\phi_{\mathrm{final}},$$
$$\phi_t := (u_{a,t}, v_{a,t}, b_{a,t}, \mathbf{z}_t), \ \mathbf{z}_t := (z_t^{(\sigma)}, z_t^{(s)}, z_t^{(x)}).$$

**Experimental Results** Per the results in Figure 1, DAL outperforms the current state-of-the-art methods in every

synthetic experiment for both choices of detectors. DAL only abandons the actions chosen by the stationary bandit algorithm and restarts learning when an efficient change detector flags a mean-shift in rewards; hence, it avoids unnecessary restarts, especially when the intervals between the change-points are long enough for such detectors to correctly flag said changes without false alarms. Regarding drifting non-stationarity, DAL significantly outperforms all other methods, faring better than both WeightUCB and ADA-OPKB, which attain the optimal regret bound in the drift setup. Finally, Figure 2 shows that DAL's empirical gains do not come at the cost of prohibitive computation: across settings, DAL remains runtime-competitive with the fastest top-performing baselines and is substantially faster than MASTER in most cases. Thus, DAL achieves the best regret performance across all synthetic experiments while retaining practical computational efficiency.

## 3.4. Real-World Experiments

To validate DAL's practical effectiveness we conduct experiments on 9 real-world datasets and the construction details are given in the Appendix.

**Micro-Prefetcher Selection.** We evaluate on a novel NS bandit benchmark constructed from the microarchitecture prefetcher dataset of Gerogiannis & Torrellas (2023).[1] The environment has $K = 11$ prefetcher configurations (actions) with rewards given by normalized IPC in $[0, 1]$ over a horizon $T = 26224$. Following Gerogiannis & Torrellas (2023), we also evaluate D-UCB (Kocsis & Szepesvári, 2006) in its native form, while all other methods are modeled as NS-SCBs. Performance is measured by cumulative reward.

**Stock Market Benchmarks.** We consider two NS-KB

---

[1]We aim to release this dataset to facilitate real-world experimentation by the bandit research community.

---

**Algorithm 2** **G**eneralized **L**ikelihood **R**atio Test

---

1: **Input**: History $\mathcal{H} = \{X_1, \ldots, X_n\}$, $\delta_{\mathrm{F}}$, $\delta_{\mathrm{D}}$, KL divergence $\mathrm{kl}(\cdot, \cdot)$
2: **for** $k = 1$ to $n - 1$ **do**
3:     Compute empirical means $\hat{\mu}_{1:k}$, $\hat{\mu}_{k+1:n}$, $\hat{\mu}_{1:n}$
4:     $\mathrm{GLR}_k \leftarrow k\,\mathrm{kl}(\hat{\mu}_{1:k}, \hat{\mu}_{1:n})$
5:                $+ (n - k)\,\mathrm{kl}(\hat{\mu}_{k+1:n}, \hat{\mu}_{1:n})$
6:     **if** $\mathrm{GLR}_k \geq \beta_{\mathrm{GLR}}(n, \delta_{\mathrm{F}})$ **then**
7:         **return** detection

---

**Algorithm 3** **G**eneralized **S**hiryaev–**R**oberts Test

---

1: **Input**: History $\mathcal{H} = \{X_1, \ldots, X_n\}$, $\delta_{\mathrm{F}}$, $\delta_{\mathrm{D}}$, KL divergence $\mathrm{kl}(\cdot, \cdot)$, $\mathrm{GSR}_k \leftarrow 0$
2: **for** $k = 1$ to $n - 1$ **do**
3:     Compute empirical means $\hat{\mu}_{1:k}$, $\hat{\mu}_{k+1:n}$, $\hat{\mu}_{1:n}$
4:     Compute $\mathrm{GLR}_k$ according to Alg. 2
5:     $\mathrm{GSR}_k \leftarrow \mathrm{GSR}_k + \exp(\mathrm{GLR}_k)$
6:     **if** $\mathrm{GSR}_k \geq \beta_{\mathrm{GSR}}(n, \delta_{\mathrm{F}})$ **then**
7:         **return** detection

---

benchmarks following Deng et al. (2022): one based on their original Quandl dataset and one constructed from stocks retrieved via Yahoo Finance.[2] In the Yahoo-based benchmark, we retain assets with sufficient history ($T = 2000$) and select the 50 most volatile as actions. Rewards are derived from daily closing prices, the empirical price covariance is used as the kernel, and we add i.i.d. $\mathcal{N}(0, 0.01)$ noise to increase difficulty. Evaluation is by dynamic regret.

**COVID-NMA Clinical Benchmark.** We construct an NS-SCB benchmark from the open COVID-NMA database (Boutron et al., 2020; 2025), modeling treatment effectiveness over time using a union endpoint on reported outcomes. Treatments are mapped into $K = 13$ classes and month-level counts are expanded and concatenated into $T \approx 7.4 \times 10^4$. Performance is evaluated by cumulative reward.

**Click Log Benchmarks.** We evaluate on the Yahoo! R6A click log dataset[3] following standard preprocessing of prior work (Cao et al., 2019b; Seznec et al., 2020), yielding an NS-SCB problem with $K = 64$ actions and horizon $T = 50000$; the metric is dynamic regret. In addition, we construct a fixed-arm replay benchmark from the Yahoo! R6B click logs as an NS-CB with 51 actions and $T = 50000$, leveraging uniform-random logging for unbiased IPS evaluation (Li et al., 2011); the metric is (replay) cumulative reward.

**Live Traffic Benchmark.** We build an NS-GLB benchmark from the Criteo live traffic dataset (Diemert et al., 2017), following Russac et al. (2019) but introducing changes via a geometric change-point model and using a horizon of $T = 50000$. Evaluation is based on dynamic regret.

**Sensor Correlation Benchmark.** We use the Bioliq sensor dataset of Komiyama et al. (2024) to construct an NS-SCB environment with $K = 190$ actions over a week-long horizon. Rewards are binary indicators derived from temporal correlation statistics following Komiyama et al. (2024). Performance is measured by cumulative reward.

---

[2]Data retrieved from Yahoo Finance using the publicly available yfinance package. Used solely for non-commercial, academic research purposes.

[3]Yahoo! Front Page Today Module User Click Log Datasets: https://webscope.sandbox.yahoo.com.

**Ad Recommendation Benchmark.** We evaluate on the Zozo ad recommendation environment (Saito et al., 2021), using the preprocessing of Komiyama et al. (2024) but retaining all $K = 80$ ads as actions. Rewards are binary click indicators and evaluation is by cumulative reward.

Based on Figure 3, DAL consistently outperforms all state-of-the-art baselines across real-world benchmarks, in both dynamic regret and cumulative reward with both GLR and GSR. We attribute this strong performance to the robustness DAL demonstrates in the synthetic settings, which capture a range of challenging NS scenarios. These findings underscore DAL's practical effectiveness. In what follows, we provide a theoretical explanation for its performance.

## 4. Theoretical Insights

### 4.1. On Effective Detection

When selecting a non-stationarity detector, accuracy and efficiency are essential for ensuring optimal regret growth. Any detector aiming to identify distribution shifts inherently requires a certain number of samples, both before and after the change. This sample complexity should scale appropriately to avoid negatively impacting the total regret. To this end, the GLR and GSR tests have been shown to achieve a pre- and post-change sample complexity of the order $\log T$ (Huang & Veeravalli, 2025). Since logarithmic terms are disregarded in regret analyses, this suggests that integrating such mechanisms could achieve optimal regret growth.

The stopping time $\tau$ of a change detector $\mathcal{D}$ denotes the time-step at which a change is identified. Let $\mathbb{P}_\nu$ and $\mathbb{E}_\nu$ be the probability and expectation with change-point at $\nu$, and $\mathbb{P}_\infty$ and $\mathbb{E}_\infty$ be the ones with no change-point. The *latency* $\ell_{\mathcal{D}}$ is the length of time post-change within which a change is declared with a probability $1 - \delta_{\mathrm{D}}$, i.e., $\ell_{\mathcal{D}}$ is defined as:

$$\inf\{t \in [T] : \mathbb{P}_\nu(\tau \geq \nu + t) \leq \delta_{\mathrm{D}}, \ \forall \nu \in [m_{\mathcal{D}} + 1, T - t]\}$$

where $m_{\mathcal{D}}$ is the length of the pre-change window at which no changes occur. A good detector seeks to minimize $\ell_{\mathcal{D}}$ while ensuring low false-alarm probability over horizon $T$, namely $\mathbb{P}_\infty(\tau \leq T) \leq \delta_{\mathrm{F}}$ with $\delta_{\mathrm{F}} \in (0, 1)$. To ensure order-optimal regret for DAL, the detector $\mathcal{D}$ must satisfy:

**Property 4.1.** $\ell_{\mathcal{D}} + m_{\mathcal{D}} = \mathcal{O}(\log T + \log(1/\delta_{\mathrm{F}}) + \log(1/\delta_{\mathrm{D}}))$.

This condition is crucial in the proof of Theorem 4.8 in the Appendix. We employ the GLR and GSR tests since they satisfy Property 4.1 (Huang & Veeravalli, 2025), with thresholds $\beta_{\mathrm{GLR}}(n, \delta_{\mathrm{F}}) = \mathcal{O}(\log(n^{3/2}/\delta_{\mathrm{F}}))$ and $\beta_{\mathrm{GSR}}(n, \delta_{\mathrm{F}}) = \mathcal{O}(n^{5/2}/\delta_{\mathrm{F}})$. In experiments, GSR performs slightly better than GLR, but the difference is minor since both satisfy Property 4.1. This shows that the good performance is due to the *design* of DAL rather than the specifics of a single detector.

The Bernoulli GLR and GSR are used for sub-Bernoulli rewards with $\mathrm{kl}(x, y) = x \ln(x/y) + (1-x) \ln\left(\frac{1-x}{1-y}\right)$, and the Gaussian variants are for $\sigma^2$-sub-Gaussian rewards with $\mathrm{kl}(x, y) = (x - y)^2/(2\sigma^2)$.

To select which samples should be fed into the detector, one needs to properly select the covering set $\mathcal{A}_{\mathrm{e}}$, so that it contains actions that can capture changes in the reward function for any context. However, changes cannot be arbitrarily small, as no change detector may be able to identify them. Hence, $\mathcal{A}_{\mathrm{e}}$ should be designed such that whenever a change occurs, reward sequences associated with at least one context-action pair in $\mathcal{C} \times \mathcal{A}_{\mathrm{e}}$ exhibit an *appreciable* mean-shift. Define

$$\Delta_{\mathrm{c}} := \inf_{f \neq f'} \max_{(c,a) \in \mathcal{C} \times \mathcal{A}_{\mathrm{e}}} |f(c,a) - f'(c,a)|.$$

Then, $\Delta_{\mathrm{c}}$ captures how well the context-action pairs in $\mathcal{C} \times \mathcal{A}_{\mathrm{e}}$ can discern between candidate reward functions. According to (Huang et al., 2025), $\Delta_{\mathrm{c}}$ crucially affects the performance of the GLR and GSR tests, as their pre- and post-change sample complexity grows with $1/\Delta_{\mathrm{c}}^2$. The more discernible the changes are, the easier the detection becomes. Since forced exploration incurs regret, $\mathcal{A}_{\mathrm{e}}$ should be chosen to minimize $N_{\mathrm{e}}$ while maximizing $\Delta_{\mathrm{c}}$. However, this cannot be done since the function $f_t$ is unknown. Hence, we provide the ways with which one can ensure appreciable mean-shift (i.e., $\Delta_{\mathrm{c}} > 0$) in settings where the reward function has a certain *structure* (e.g., linear dependence on the arms or prescribed smoothness). Specifically, the NS-PB and NS-KB settings satisfy such conditions. Using these choices of $\mathcal{A}_{\mathrm{e}}$, one can guarantee order-optimal regret in certain cases, as shown in the next section. The proofs of the following propositions are given in the Appendix.

**Proposition 4.2.** *In NS-PBs, $\mathcal{A}_{\mathrm{e}}$ can be any arbitrary maximal linearly independent subset of $\mathcal{A}$.*

**Proposition 4.3.** *In NS-KBs, assume that $\mathcal{A} \subseteq [0, R]^d$ w.l.o.g., and that there exists an $\tilde{a} \in \mathcal{A}$ s.t.*

$$\inf_{f \neq f'} |f(\tilde{a}) - f'(\tilde{a})| > L_T,$$

*for some $L_T > 0$. Let $\delta_T := L_T/(2BL_u)$, where $BL_u$ is the Lipschitz constant of all $f \in H_k(\mathcal{A})$ and let $\mathcal{V}_T \subset \mathcal{A}$ be the set of the centers of the balls of an arbitrary $\delta_T$-cover. Then, $\mathcal{A}_{\mathrm{e}}$ can be taken as $\mathcal{V}_T$, with $|\mathcal{V}_T| \leq \lceil\sqrt{d}R/2\delta_T\rceil^d = \lceil\sqrt{d}BL_u R/L_T\rceil^d$.*

**Remark 4.4.** *In NS-CBs, if $f_t$ and $\mathcal{A}$ satisfy the structural assumptions of the preceding propositions for any fixed context, we can set $\mathcal{A}_{\mathrm{e}}$ similarly. Without such structure, we set $\mathcal{A}_{\mathrm{e}} = \mathcal{A}$, as $\mathcal{A}$ is finite.*

The constructions in Propositions 4.2, 4.3 and Remark 4.4 are one-time preprocessing steps performed before DAL is run, and do not introduce per-round overhead. In the PB case, the cost scales with the number of inspected candidate actions and the corresponding linear-independence checks, rather than with the horizon $T$. The purpose of this construction is structural: once $\mathcal{A}_{\mathrm{e}}$ spans the relevant action space, any change in the underlying parameter must induce a reward-mean change on at least one monitored action. Hence, the preprocessing cost is typically negligible compared with the online learning cost, while ensuring that DAL monitors a sufficiently informative subset of actions.

## 4.2. On Order-Optimality in Piecewise Stationary Environments

In the PS setting, the minimax regret lower bound under bandit feedback is $\tilde{\Omega}(\sqrt{N_T T})$ (Garivier & Moulines, 2011),[4] which applies across all settings considered in this work, differing only in problem-dependent constants. Under certain conditions on the minimum spacing between change-points, our algorithm matches this bound with state-of-the-art dependence on these constants. Specifically, the assumption states that $\nu_k - \nu_{k-1}$ should be large enough to acquire enough samples to trigger restarts. For brevity, we first define the relevant quantities and then state the assumption.

**Definition 4.5.** For PS-PBs and PS-KBs, let $m_k := \lceil N_{\mathrm{e}}/\alpha_k \rceil m_{\mathcal{D}}$ and $\ell_k := \lceil N_{\mathrm{e}}/\alpha_k \rceil \ell_{\mathcal{D}}$ for $k \in [N_T]$. For PS-CBs, let $m_k := \lceil N_{\mathrm{e}}/\alpha_k \rceil \lceil m_{\mathcal{D}}/s + \log T/4s^2 + \sqrt{m_{\mathcal{D}} \log(T)/2s^3} + (\log T)^2/16s^4 \rceil$ and $\ell_k := \lceil N_{\mathrm{e}}/\alpha_k \rceil \lceil \ell_{\mathcal{D}}/s + \log(T)/4s^2 + \sqrt{\ell_{\mathcal{D}} \log T/2s^3} + (\log T)^2/16s^4 \rceil$ for $k \in [N_T]$, with $s := \min_{c \in \mathcal{C}, t \in [T]: \mathcal{P}_t(c) > 0} \mathcal{P}_t(c)$.

**Assumption 4.6.** Assume $\nu_1 \geq m_1$ and $\nu_k - \nu_{k-1} \geq \ell_{k-1} + m_k$ for $k \in \{2, \ldots, N_T\}$.

In PS-PBs and PS-KBs, DAL performs round-robin forced exploration on each arm every $\lceil N_{\mathrm{e}}/\alpha_k \rceil$ rounds. Thus, the scaling of $m_{\mathcal{D}}$ and $\ell_{\mathcal{D}}$ in Definition 4.5 is necessary for Assumption 4.6 to guarantee that the change detector in each arm observes at least $m_{\mathcal{D}}$ pre-change samples and $\ell_{\mathcal{D}}$ post-change samples. In PS-CBs, each context–action pair is only seen in expectation (not deterministically) at least once every $\lceil N_{\mathrm{e}}/\alpha_k \rceil/s$ rounds due to randomness. Thus, in Assumption 4.6, we increase the change-point separation, as

---

[4]We use $\tilde{\Omega}(\cdot)$ to hide polylogarithmic factors.

*Table 1.* Regret comparison for PS bandits, under Assumption 4.6. DAL seamlessly transfers the order-wise dependence from the stationary setting to the PS setting, while attaining or improving the state-of-the-art performance bounds in all bandit settings.

| Setting | State-of-the-art PS regret | DAL PS regret | DAL's Input Stationary Regret |
|---|---|---|---|
| PS-LBs | $\tilde{\mathcal{O}}\big(d\sqrt{N_T T}\big)$ | $\tilde{\mathcal{O}}\big(d\sqrt{N_T T}\big)$ | $\tilde{\mathcal{O}}\big(d\sqrt{T}\big)$ |
| PS-GLBs | $\tilde{\mathcal{O}}\big(d\sqrt{N_T T}\big)$ | $\tilde{\mathcal{O}}\big(d\sqrt{N_T T}\big)$ | $\tilde{\mathcal{O}}\big(d\sqrt{T}\big)$ |
| PS-CBs | $\tilde{\mathcal{O}}\Big(\sqrt{|\mathcal{A}|\,N_T T \log|\Pi|}\Big)$ | $\tilde{\mathcal{O}}\Big(\sqrt{|\mathcal{A}|\,N_T T \log|\Pi|}\Big)$ | $\tilde{\mathcal{O}}\Big(\sqrt{|\mathcal{A}|\,T \log|\Pi|}\Big)$ |
| PS-SCBs | $\tilde{\mathcal{O}}\Big(d^{2/3}T^{2/3}N_T^{1/3}\Big)$ | $\tilde{\mathcal{O}}\big(d\sqrt{N_T T}\big)$ | $\tilde{\mathcal{O}}\big(d\sqrt{T}\big)$ |
| PS-KBs | $\tilde{\mathcal{O}}\big(\sqrt{d\,\gamma_T\,N_T T}\big)$ | $\tilde{\mathcal{O}}\big(\sqrt{\gamma_T\,N_T T}\big)$ | $\tilde{\mathcal{O}}\big(\sqrt{\gamma_T T}\big)$ |

shown in Definition 4.5, to collect the $m_{\mathcal{D}}$ and $\ell_{\mathcal{D}}$ samples with high probability. These conditions allow $\mathcal{D}$ to reliably detect a change (Property 4.1).

The assumption on the minimum separation between change-points essentially requires scaling as $\tilde{\mathcal{O}}(\sqrt{T/k})$. However, this condition primarily emerges from a conservative proof technique, where missed detections are aggregated into a single adverse event. Practically, and as corroborated by our experiments, this assumption is often violated without negatively impacting the regret performance—even under scenarios with frequent and arbitrarily placed change-points (e.g. $\xi = 0.4$). We suspect that this resilience arises because any detector satisfying Property 4.1, while potentially missing isolated short intervals, reliably detects subsequent changes when stationary segments exceed the threshold length. Even if a change is entirely missed during a segment shorter than $\tilde{\mathcal{O}}(\sqrt{T/k})$, the resulting regret remains under that order. Conversely, when the assumption holds, optimal regret is provably guaranteed. Thus, the required separation threshold acts as a practical "sweet spot": segments longer than this threshold are detected reliably, ensuring optimal performance, while shorter segments incur minimal regret, thereby preserving overall optimal regret.

**Remark 4.7.** *Assumption 4.6 is necessary to prove the order-optimality, **but it is not for practical performance**. None of our experiments enforced this assumption, and DAL dominated in both the synthetic and the real-world simulations as shown in Section 3.*

Based on Algorithm 1, DAL can incorporate any stationary bandit algorithm. Since different algorithms yield different regret guarantees, DAL attains order-optimal regret in PS environments only when the stationary component has optimal minimax regret, namely $\tilde{\mathcal{O}}(d\sqrt{T})$ in PBs, $\tilde{\mathcal{O}}(\sqrt{\gamma_T T})$ in KBs, and $\tilde{\mathcal{O}}(\sqrt{|\mathcal{A}|\log|\Pi|\,T})$ in CBs. This requirement is formalized in Theorem 4.8.

Thus, to characterize DAL's performance under piecewise stationarity, we employ the methodology of (Huang et al., 2025), incorporating the regret analysis of the stationary bandit algorithm and that of the change detector. Since we

are studying general bandits, additional novel analysis is required. Due to space constraints, the full analysis and proof of Theorem 4.8 are deferred to the Appendix.

**Theorem 4.8.** *For the PS setting, consider DAL with a detector $\mathcal{D}$ that satisfies Property 4.1, a stationary bandit algorithm $\mathcal{B}$ with regret upper bound $R_{\mathcal{B}}$ concave and increasing with $T$, a covering set $\mathcal{A}_{\mathrm{e}}$ and forced exploration frequencies $(\alpha_k)_{k=1}^T$. If Assumption 4.6 holds, $\alpha_k = \sqrt{k|\mathcal{C}|N_{\mathrm{e}}}/(2\sqrt{T}\log^2 T)$, $\delta_{\mathrm{F}} = \delta_{\mathrm{D}} = T^{-\eta}$, with $\eta > 1$, and $R_{\mathcal{B}}(T) = \tilde{\mathcal{O}}(d^p \gamma_T^q (|\mathcal{A}|\log|\Pi|)^r \sqrt{T})$ with $p, q, r \geq 0$, then DAL's regret satisfies $R_T = \tilde{\mathcal{O}}(d^p \gamma_T^q (|\mathcal{A}|\log|\Pi|)^r \sqrt{N_T T} + \sqrt{|\mathcal{C}|N_{\mathrm{e}} N_T T})$.*

Using Theorem 4.8, Propositions 4.2, 4.3 and Remark 4.4 we present the optimal regret of DAL.

**Corollary 4.9.** *Assume that the conditions of Theorem 4.8 hold. In parametric bandits, select $\mathcal{A}_{\mathrm{e}}$ as in Proposition 4.2. In kernelized bandits, select $\mathcal{A}_{\mathrm{e}}$ as in Proposition 4.3 with $\delta_T := \frac{Rd^{1/2-2p/d}}{2(C\gamma_T^{2q})^{1/d}}$ for some $C > 0$. In contextual bandits, set $\mathcal{A}_{\mathrm{e}}$ as in Remark 4.4. Then, DAL attains*

$$R_T = \tilde{\mathcal{O}}(d^p \gamma_T^q (|\mathcal{A}|\log|\Pi|)^r \sqrt{N_T T}).$$

*That is, if the stationary algorithm has order-optimal regret, DAL retains optimality. This also holds when $N_{\mathrm{e}} < d$ in PS-PBs, when $N_{\mathrm{e}} < \gamma_T$ in PS-KBs, and when $\Pi$ is the universal set of all mappings from $\mathcal{C}$ to $\mathcal{A}$.*

**State-of-the-art Regret.** Corollary 4.9 yields regret bounds across all considered settings while allowing flexible choices of the stationary algorithm. Choosing the stationary algorithms in Section 3, which achieve optimal stationary regret, leads to the bounds summarized in Table 1. For self-concordant bandits, the strongest known bound is prior-based (Wang et al., 2023); although (Russac et al., 2021) attains a sharper rate, its analysis relies on substantially stronger assumptions than (Wang et al., 2023).[5] Table 1 highlights the important property of DAL: the order-wise de-

---

[5] While MASTER may be extendable to PS-SCBs, no corresponding regret bound is currently known.

pendence on problem parameters from the stationary setting seamlessly transfers to the PS setting without degradation.

### 4.3. On Drifting Environments

Based on the previous section, at first glance, one can expect that DAL is not able to handle drifting non-stationarity. Our results in Section 3 naturally lead us to ask when and why DAL performs well in drifting environments. As a first step to study this, we perform another experiment with LBs. Specifically, the parameter $\theta_t$ in each time-step $t$ evolves randomly as, $\quad \theta_{t+1} := \theta_t + \zeta_{t+1}$

where $\zeta_{t+1} \in \mathbb{R}^d$ is chosen uniformly over a $\delta$-ball. If the resulting $\theta_{t+1}$ violates the norm-bound $S$, we disregard that choice of $\zeta_{t+1}$ and sample again. We sample $\varepsilon_t \sim \mathcal{N}(0, 0.1)$ at each $t$. We compare the cumulative dynamic regret up to time $T$ of DAL+LinUCB with GLR, and WeightUCB over a range of $\delta$'s in Figure 4. The remaining parameters are chosen to be the same as those in Section 3, with the exception of $d = 5$. The DAL algorithm performs better than WeightUCB for smaller values of $\delta$, but the conclusion reverses upon increasing $\delta$.

We now shed light on our hypothesis behind the observations from Figure 4. For playing an action $a \in \mathcal{A}$ at time $t + 1$, we get the random reward,

$$X_{t+1} = \langle \theta_t, a \rangle + \langle \zeta_{t+1}, a \rangle + \varepsilon_{t+1}.$$

If the $\theta_t$ does not change, then the reward from playing action $a$ would have been $X'_{t+1} = \langle \theta_t, a \rangle + \varepsilon'_{t+1}$, where $\varepsilon'_{t+1}$ is another realization of the noise. Statistically, a specific instance of $\langle \zeta_{t+1}, a \rangle + \varepsilon_{t+1}$ and $\varepsilon'_{t+1}$ are close to each other, when $\delta$ is small, albeit the resulting (small) mean-shift due to the drift in the governing parameter. For practical purposes, the impact of the drift can be absorbed into the noise term $\varepsilon_{t+1}$ when $\delta$ is small. As a result, one expects an algorithm tailored to handle piecewise stationarity to perform reasonably well for slowly drifting environments. Conversely, if $\delta$ is large, the bias induced by $\zeta_{t+1}$ is large enough to disallow absorbing it into the noise term. Over a few time-steps, the cumulative effect of this compounding bias is then large enough to completely violate the stationarity assumption. With large enough $\delta$, the change in $\theta_t$ over a few time-steps can be considered large enough for a restart.

## 5. Summary and Outlook

We introduced DAL, a practical, prior-free black-box framework for general PS bandits. Its plug-and-play design integrates seamlessly with a wide range of stationary bandit algorithms and different detectors. Through extensive experiments in both PS and drifting settings, spanning synthetic and real-world benchmarks, DAL consistently outperforms all prior-free baselines, including the black-box

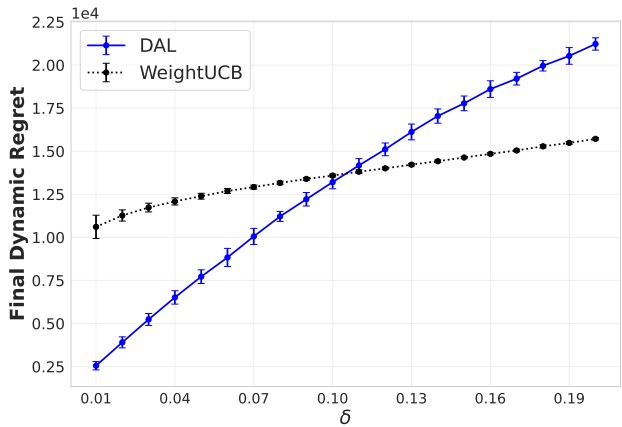

*Figure 4.* Final dynamic regret vs. radius $\delta$: Drifting LBs. Smaller $\delta$ allows the variation to be absorbed into the noise, enabling DAL to outperform WeightUCB.

gold standard MASTER and the state-of-the-art methods ADA-OPKB and ADA-ILTCB+, and even surpasses leading prior-based methods like WeightUCB and WGP-UCB. Its leading performance in real-world scenarios highlights its value as a practical and effective solution.

On the theoretical side, using existing results and providing novel techniques, we showed that DAL inherits and adapts the regret guarantees of its stationary input algorithm, achieving order-optimal regret under piecewise stationarity, with mild change-point separation. As a result, it matches the best existing bounds in PS-LBs, PS-GLBs and PS-CBs while improving the best known bounds for PS-SCBs and PS-KBs. Regarding drifting non-stationarity, we hypothesized key conditions under which DAL excels–an insight further validated through additional experiments under drifting settings. Our results suggest that a well-designed algorithm for the PS setting can extend to a broad range of drifting scenarios, bridging the gap between these two regimes.

While DAL advances both theory and practice, it opens new directions. First, regret guarantees for detection-based methods in drifting environments remain unexplored. Second, the current regret bounds for DAL rely on a separation condition between change-points—a standard assumption in the detection-based literature (see e.g., Besson et al. (2022); Huang et al. (2025)), which nonetheless limits the extent to which DAL achieves fully prior-free theoretical optimality. Addressing these gaps would deepen our understanding of detection-based methods in more continuous forms of non-stationarity. Finally, DAL's modular nature invites extensions to broader settings, including reinforcement learning. We believe that deepening the study of piecewise stationarity may be the key to tackling these broader challenges and DAL can serve as a solid foundation towards that goal.

## Acknowledgments

AG gratefully acknowledges Gerasimos Gerogiannis for sharing the microarchitecture data used to construct the real-world benchmark. The authors thank the anonymous reviewers for their feedback. This work was supported in part by a grant from the C3.ai Digital Transformation Institute, and in part by the Army Research Laboratory under Cooperative Agreement W911NF-17-2-0196, through the University of Illinois at Urbana-Champaign.

## Impact Statement

This paper presents work whose goal is to advance the field of machine learning. There are many potential societal consequences of our work, none of which we feel must be specifically highlighted here.

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

# A. Experimental Details

## A.1. On Forced Exploration in Finite Action Spaces

**Covering Set Construction.** In practice, the covering set $\mathcal{A}_e$ is selected according to Propositions 4.2, 4.3, and Remark 4.4 together with the specifications of Corollary 4.9. However, in finite-action settings, the full construction may not be feasible: the action set $\mathcal{A}$ may not contain enough elements to satisfy the required conditions. For instance, in the NS-PB setting, $\mathcal{A}$ may not include $d$ linearly independent actions, while in the NS-KB case, it may lack a full $\delta_T$-covering net for the chosen $\delta_T$ in Corollary 4.9. One might expect that when $|\mathcal{A}_e| < d$ in PS-PBs or $|\mathcal{A}_e| < \gamma_T$ in PS-KBs, the inability to detect all possible changes would degrade DAL's performance. In practice, however, DAL does not need to restart when changes in the reward function leave the mean reward of each action unchanged. Crucially, as discussed in Appendix B.6, DAL retains order-optimality even in these constrained regimes. Accordingly, whenever $|\mathcal{A}| < d$ or $|\mathcal{A}| < \gamma_T$, we simply set $\mathcal{A}_e = \mathcal{A}$. In our experiments, the action set is finite (typically in the hundreds). For PS-PBs, the random generation of actions almost always guarantees $d$ linearly independent vectors. For PS-KBs, since $\gamma_T$ is typically large, we also use the full action set $\mathcal{A}$ as $\mathcal{A}_e$ without impacting performance. On the other hand, since the regret bounds in PS-CBs include $|\mathcal{A}|$, as it is finite, in any PS-CB setting we can simply set $\mathcal{A}_e = \mathcal{A}$.

**Practical Implementations.** For NS-PBs, we construct $\mathcal{A}_e$ by greedily selecting linearly independent actions until we obtain $d$ such vectors, where $d$ is the dimension of the action space. In the NS-KB setting, $\mathcal{A}_e$ is formed by building a $\delta_T$-cover over the bounded action space and choosing the centers of the covering balls. If the action space is continuous and bounded, these centers suffice to cover the space. If the action space is finite and $N_e < d^{2p}\gamma_T^{2q}$, then the entire set $\mathcal{A}$ serves as the covering set, as established in Corollary 4.9. Otherwise, if $N_e > d^{2p}\gamma_T^{2q}$, we select the $d^{2p}\gamma_T^{2q}$ actions closest to the covering-ball centers. Finally, in the NS-CB setting, selecting a smaller $\mathcal{A}_e$ compared to $\mathcal{A}$ does not affect regret, but improves practical performance due to less forced exploration. Thus, depending on the reward function and action set structures, it is recommended to decrease the cardinality of $\mathcal{A}_e$ as much as possible.

**Sensitivity of $\mathcal{A}_e$** As shown in Algorithm 1, DAL's forced exploration depends on $N_e$, the cardinality of $\mathcal{A}_e$. Intuitively, a larger $N_e$ increases the exploration burden, since DAL must select more actions to detect changes. In all cases, DAL limits the cardinality of $\mathcal{A}_e$ to the minimum number of actions needed to characterize the reward function for detection and learning. These cardinalities match the quantities appearing in the minimax stationary regret bounds (e.g., $d$ for PBs, $\gamma_T$ for KBs, and $|\mathcal{A}|$ for CBs). This principle guided our design of the covering-set selection procedures.

- **NS-PBs.** In the NS-PB setting, Proposition 4.2 shows that the cardinality of a suitable covering set is at most $d$. Thus, even if the underlying action space is infinite, DAL only needs to explore at most $d$ actions in $\mathcal{A}_e$. In this sense, DAL is not sensitive to the size of the continuous action space: it pays only a $d$-dependent cost. If there are multiple choices of $d$ linearly independent actions, the practical performance depends on the induced change magnitude $\Delta_c$ (as discussed in Section 4.1). For a fixed non-stationarity model, some choices of $d$ actions may yield larger $\Delta_c$, improving pre- and post-change sample complexity. However, our regret analysis accounts for the worst case over $\Delta_c$, so, at the theoretical level, DAL is not sensitive to which particular $d$ actions are chosen.

- **NS-KBs.** In the NS-KB setting, the sensitivity of DAL is governed by the smoothness of the RKHS. If the Lipschitz constant $BL_u$ is small, the RKHS contains smooth functions, so we can use a relatively large $\delta_T$, leading to a smaller covering set $\mathcal{A}_e$ (and thus a smaller $N_e$). If the RKHS contains less smooth functions (larger $BL_u$), we require a smaller $\delta_T$ to detect changes reliably, which increases $N_e$. Nevertheless, to attain order-optimality DAL only needs to explore at most $\gamma_T$ actions, which is finite and significantly smaller than the (possibly infinite) original action space. As in NS-PBs, DAL is more sensitive to the underlying function class (smoothness) than to the raw size of the continuous action space.

- **NS-CBs.** In the NS-CB case, the action set is finite, and one must fully explore all actions in order to characterize changes in the reward function, since the rewards can be completely uninformative about structural properties beyond their realized values.

**Experimental Choices.** In our experiments, for NS-PBs the action set is sampled from a multivariate Gaussian distribution, which ensures the existence of $d$ linearly independent actions. Thus, we always set $N_e = d$ using the greedy selection procedure described above. For NS-KBs, the regret bound for $N_e$ obtained from Theorem 4.8 and Corollary 4.9 is extremely

large for our horizons, implying that $|\mathcal{A}| < \gamma_T$. Consequently, in all NS-KB experiments we simply take $\mathcal{A}_e = \mathcal{A}$ and set $N_e$ equal to the number of available actions, which yielded optimal performance. Finally, since the reward does not exhibit any structure with the arms in PS-CBs, we simply set $\mathcal{A} = \mathcal{A}_e$.

### A.2. Real-World Data Preprocessing

**Microarchitecture Prefetcher Selection Benchmark.** We introduce a non–stationary bandit dataset derived from the MICRO'23 study of Gerogiannis & Torrellas (2023), built on the SPEC06/17 benchmark suites. Each action corresponds to one of 11 L2 prefetcher configurations (next–line on/off, stream degree, stride degree). The sequence spans $T{=}26224$ rounds; at round $t$, the reward is the trace–level normalized instructions–per–cycle in $[0, 1]$, computed from performance counters. We obtained the data directly from the original authors, and note that reproducing the exact series from scratch is not feasible without the same stack, microarchitectural parameters, and arm schedules described in the paper. We aim to release the dataset to facilitate real-world experimentation by the bandit research community.

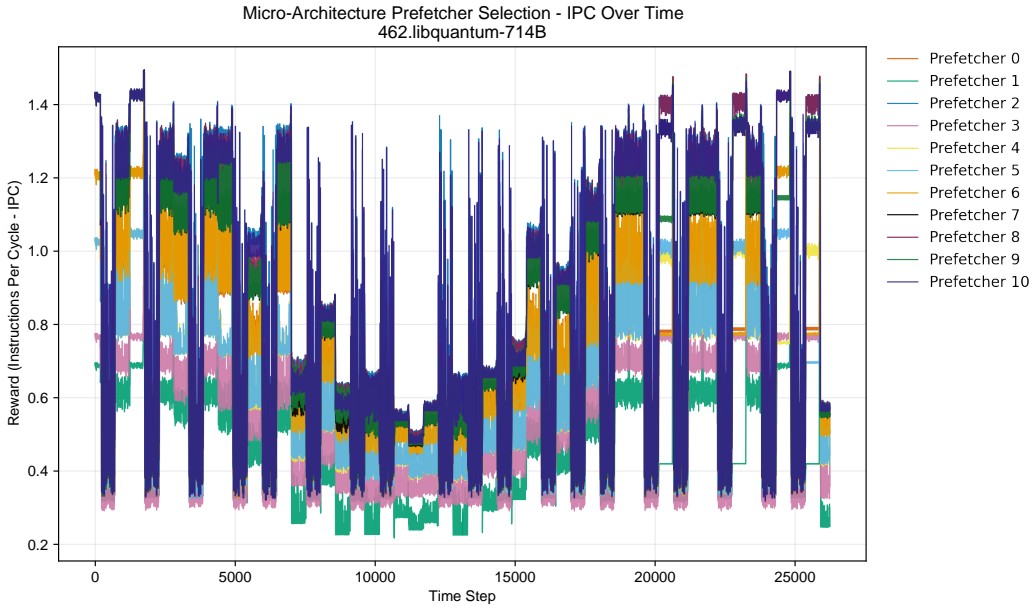

*Figure 5.* IPC of the prefetchers of the dataset over time.

**Stock Market Data Construction.** Regarding the stock market experiments we follow the procedure of (Deng et al., 2022). For the first experiment, we use the data provided in (Deng et al., 2022). For the other experiment, we collect daily closing prices of NASDAQ-100 companies using the Yahoo Finance API.[6] We filter out stocks with fewer than $T = 2000$ trading days and align all time series over the most recent $T$ dates. From this pool, we remove stocks with extremely high volatility or mean price to make the problem non-trivial, then select the top $K$ most volatile stocks from the remainder. In both cases, the stock prices are scaled accordingly to lie in $[0, 1]$. Each selected company's scaled closing price series defines the mean-reward sequence for one arm in a $K$-armed bandit problem. Finally, we corrupt the reward at each time step with $\mathcal{N}(0, 0.01)$ noise.

---

[6]Data retrieved from Yahoo Finance using the publicly available `yfinance` package. Used solely for non-commercial, academic research purposes.

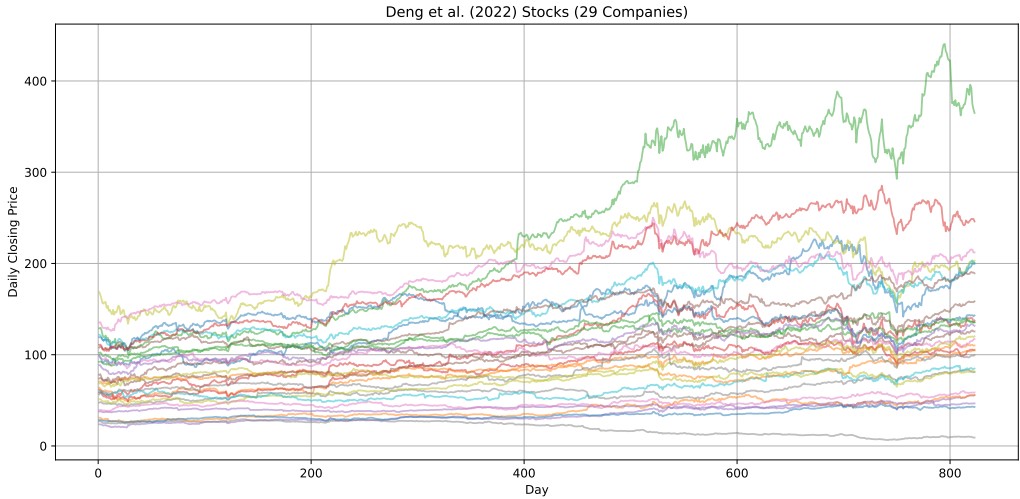

*Figure 6.* Daily closing prices from the dataset of (Deng et al., 2022).

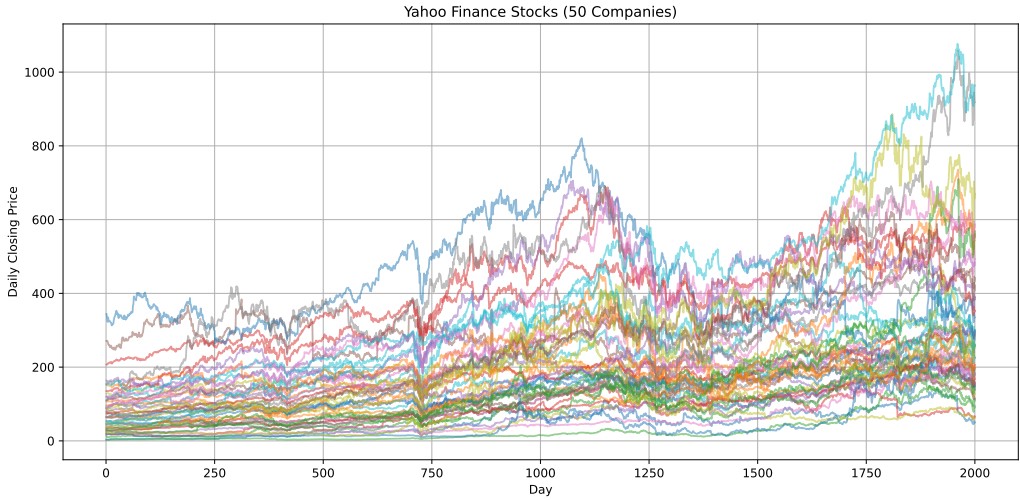

*Figure 7.* Daily closing prices obtained from Yahoo Finance.

**COVID–NMA Clinical Dataset Construction.** For the clinical benchmark based on the public COVID-NMA pharmaco-logical RCT database (Boutron et al., 2020; 2025),[7]. We use only released arm-level counts and metadata and discretize time into calendar months, assigning each trial arm to its `Start_date` (falling back to `Pub_date_online`); rows with invalid or missing dates are discarded. We deterministically map case-insensitive rules on treatment type into 13 actions: *Antivirals (any)*, *Anti–inflammatory (steroids/NSAIDs)*, *Interleukin inhibitors*, *Monoclonal antibodies (other)*, *Immunoglob-ulins/Plasma*, *Antithrombotics*, *Antimicrobials*, *Immunomodulators (non–steroid)*, *Kinase inhibitors*, *Metabolic agents*, *Supportive care*, *Control/Standard care*, and *Other/Unknown*. At the bucket–month level we compute two endpoints: (i) *Clinical Improvement @ D28* (successes = number improved; trials = reported denominator, or baseline $N$ if missing) and (ii) *Survival @ D28* derived from mortality (successes = denominator − deaths). To form a long non–stationary sequence, we adopt a union construction: for each $(k, t, \text{endpoint})$ bin we emit exactly $s_{k,t}$ ones and $n_{k,t} - s_{k,t}$ zeros and concatenate all bins in a fixed order (month, `clinD28`, `mortD28`, bucket). The sequence is fully deterministic; in our run it comprises $T \approx 7.4 \times 10^4$ rounds with 13 actions.

---

[7]Data available at: https://doi.org/10.5281/zenodo.14965887

*Figure 8.* Raw rewards for COVID-NMA Clinical dataset (Boutron et al., 2025).

**Yahoo! R6A Dataset Construction.** For the NS bandit benchmark based on the Yahoo! R6A click log dataset[8], we follow the main procedure provided in (Cao et al., 2019a; Zhou et al., 2020). We merge ten consecutive days of logs and we group the data by article ID and compute smoothed click-through rates (CTRs) using centered rolling averages over a 100-round window. This generates a time series of empirical CTRs for each article. We segment the dataset into ten distinct subperiods (each spanning half a day), filtering out actions with missing data or high noise. We further select a set of common actions present in all segments to ensure consistent tracking. We average CTRs within each subperiod and smoothing small deviations below a threshold 0.005. We stack selected actions across multiple days into a single $K \times T$ matrix, where $K$ is the number of valid actions and $T$ the compressed time horizon. To reduce spurious noise and compress the time scale, we apply local smoothing. Finally, we apply post-processing filters to remove (i) globally high-value actions (outliers with inflated CTRs), and (ii) actions that persist as best for too many segments.

**Yahoo! R6B Dataset Construction.** We follow a two-stage pipeline tailored to the Yahoo! R6B logs.[8] *Stage 1 (action vocabulary):* we scan the logs to count displays and clicks per article and select the top items using the click-through rate with a minimum display threshold of 2, yielding a fixed action set with mapping $\mathrm{id} \mapsto k \in \{0, \ldots, K-1\}$ with $K = 51$, chosen on the same window as the evaluation files. *Stage 2 (replay log):* we reprocess the files and, for each round $t$, form a feature vector $\mathbf{x}_t$ from the given features, restrict the candidate set to the Top–$K$ vocabulary to obtain $\mathcal{A}_t$, locate the displayed item's index $j_t^\star \in \{0, \ldots, |\mathcal{A}_t| - 1\}$, and record the binary click $X_t \in \{0, 1\}$; we drop rounds where the displayed item lies outside Top–$K$ or $|\mathcal{A}_t| < 2$. To increase coverage at a fixed horizon $T = 50000$, days are merged in a round-robin fashion before truncation. The resulting dataset stores $\{\mathbf{x}_t, \mathcal{A}_t, j_t^\star, r_t, t_t\}_{t=1}^T$. For offline *replay* evaluation, a policy $\pi$ observes $(\mathbf{x}_t, \mathcal{A}_t)$ and proposes $A_t \in \{0, \ldots, |\mathcal{A}_t| - 1\}$; we credit the outcome only when $a_t = j_t^\star$, and report cumulative reward $C_T = \sum_{t=1}^T \mathbb{1}\{a_t = j_t^\star\} r_t$.

[8]Yahoo! Front Page Today Module User Click Log Dataset: https://webscope.sandbox.yahoo.com.

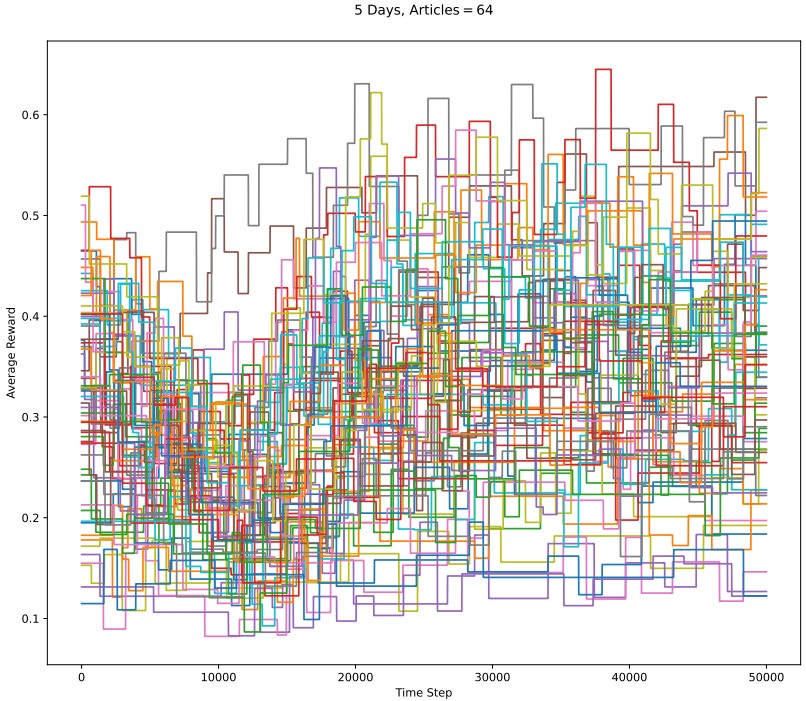

*Figure 9.* Mean rewards for the Yahoo! R6A dataset.

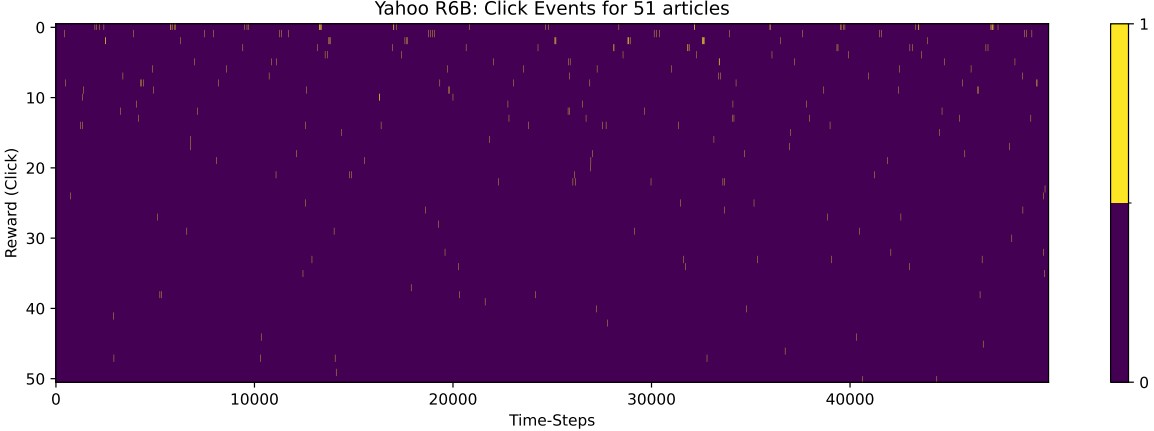

*Figure 10.* Rewards for the Yahoo! R6B dataset.

**Sensor Correlation Data Construction.** We use the Bioliq dataset from Komiyama et al. (2024), comprising a week of readings from 20 power plant sensors. Following their setup, we construct an NS-SCB environment with 190 actions: the reward is 1 if the last 1000 measurements exceed 2.04, and 0 otherwise. Evaluation is based on cumulative reward. Data available at `https://github.com/edouardfouche/G-NS-MAB/tree/master/data`.

**Ad Recommendation Data Construction.** We evaluate on the Zozo environment, a real-world ad recommender system from Saito et al. (2021), using the preprocessed dataset of Komiyama et al. (2024). We construct an NS-GLB environment with all 80 ads (unlike their 10-action setup), assigning reward 1 to any ad clicked within one second, and 0 otherwise.

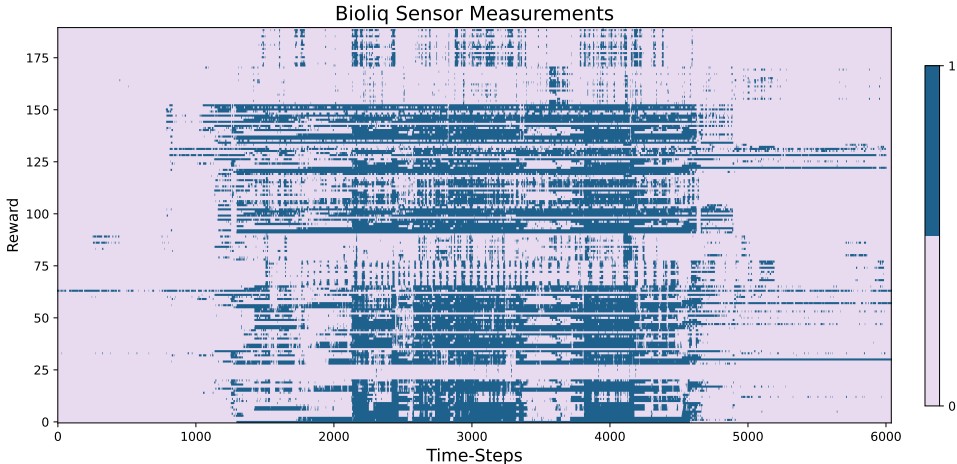

*Figure 11.* Raw rewards obtained from the Bioliq dataset (Komiyama et al., 2024).

Evaluation is based on cumulative reward. Data available at `https://github.com/edouardfouche/G-NS-MAB/tree/master/data`.

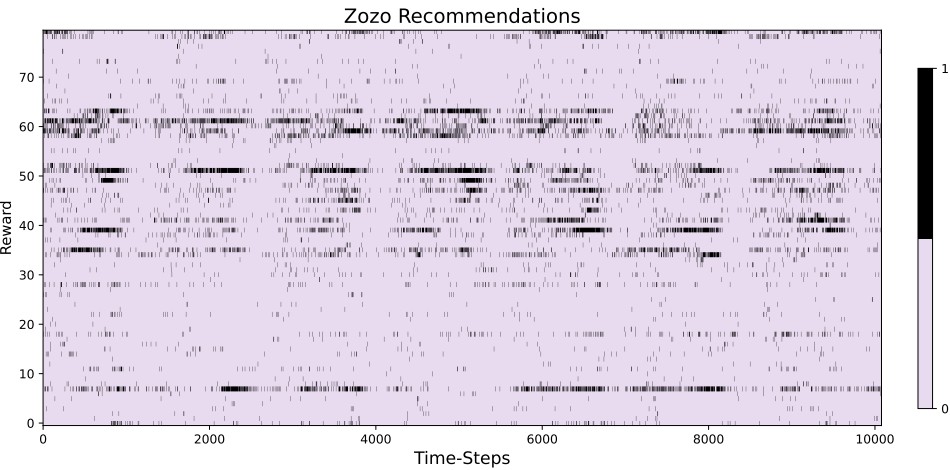

*Figure 12.* Raw rewards obtained from the Zozo dataset (Komiyama et al., 2024).

**Live Traffic Data Construction.** We construct a NS bandit environment based on the Criteo live traffic dataset (Diemert et al., 2017), following the preprocessing approach of (Russac et al., 2019) but modeling the problem as an NS-GLB rather than an NS-LB. Specifically, the dataset includes banners shown to users, associated contextual variables, and whether each banner was clicked. We retain the categorical variables `cat1` through `cat9`, along with `campaign`, which uniquely identifies each campaign. These categorical features are one-hot encoded, and a dimensionality reduction via Singular Value Decomposition selects 50 resulting features. The parameter vector $\theta^\star$ is estimated using logistic regression. Rewards are then generated from this regression model with added Gaussian noise of variance $\sigma^2 = 0.01$. Unlike (Russac et al., 2019), in which the authors employ a single change, we introduce shifts in $\theta^*$ via a geometric change-point model with parameter $\xi = 0.8$, changing $60\%$ of the $\theta^*$ coordinates at each time-step to $-\theta^*$ and extend the horizon to $T = 50000$.

### A.3. Hardware Specifications

All experiments were employed on a desktop using an Intel(R) Xeon(R) W-2245 processor with 32 GB RAM. Each experiment had a total runtime below one hour.

# B. Theoretical Results

## B.1. General Formulations of GLR and GSR

For completeness we provide the general mathematical forms of the Generalized Likelihood Ratio (GLR) and the Generalized Shiryaev-Roberts (GSR) tests. Specifically, the GLR test declares a change at time-step $\tau$, such that,

$$\tau := \inf \{n \in \mathbb{N} : G_n \geq \beta(n, \delta_{\mathrm{F}})\}$$

where the GLR statistics $G_n$ is

$$G_n := \sup_{t \in [n]} \log \left( \frac{\sup_{\theta_0 \in \mathbb{R}} \sup_{\theta_1 \in \mathbb{R}} \prod_{i=1}^{t} f_{\theta_0}(X_i) \prod_{i=t+1}^{n} f_{\theta_1}(X_i)}{\sup_{\theta \in \mathbb{R}} \prod_{i=1}^{n} f_{\theta}(X_i)} \right),$$

while the GSR test declares a change at,

$$\tau := \inf \{n \in \mathbb{N} : \log W_n \geq \beta(n, \delta_{\mathrm{F}}) + \log n\}$$

and the GSR statistic $W_n$ is given by

$$W_n := \frac{1}{n} \sum_{t=1}^{n} \left( \frac{\sup_{\theta_0 \in \mathbb{R}} \sup_{\theta_1 \in \mathbb{R}} \prod_{i=1}^{t} f_{\theta_0}(X_i) \prod_{i=t+1}^{n} f_{\theta_1}(X_i)}{\sup_{\theta \in \mathbb{R}} \prod_{i=1}^{n} f_{\theta}(X_i)} \right).$$

For both cases, $f_\theta$ can be the density of a Gaussian random variable with mean $\theta\sigma^2$ and variance $\sigma^2$ or the density of a Bernoulli random variable with the same mean. Finally, in the general case, we have that for any false alarm probability $\delta_{\mathrm{F}} \in (0,1)$, the threshold is given by

$$\beta(n, \delta_{\mathrm{F}}) = 6 \log(1 + \log(n)) + \frac{5}{2} \log \left( \frac{4n^{3/2}}{\delta_{\mathrm{F}}} \right) + 11.$$

Finally, for the practical implementation of the GLR and GSR in Algorithms 2 and 3, as per Besson et al. (2022); Huang et al. (2025) we have that, for any $n \in \mathbb{N}$ and any $t \in \{1, \ldots, n\}$:

$$\log \left( \frac{\sup_{\theta_0 \in \mathbb{R}} \prod_{i=1}^{t} f_{\theta_0}(X_i) \sup_{\theta_1 \in \mathbb{R}} \prod_{i=t+1}^{n} f_{\theta_1}(X_i)}{\sup_{\theta \in \mathbb{R}} \prod_{i=1}^{n} f_{\theta}(X_i)} \right)$$
$$= t \mathrm{kl}(\hat{\mu}_{1:t}; \hat{\mu}_{1:n}) + (n - t) \mathrm{kl}(\hat{\mu}_{t+1:n}; \hat{\mu}_{1:n})$$

where $\hat{\mu}_{t_1:t_2}$ denotes the empirical mean of the reward samples from sample $X_{t_1}$ to sample $X_{t_2}$ with $t_1 < t_2$ and $\mathrm{kl}(x; y)$ is KL-divergence between two Gaussian or Bernoulli distributions, depending on the rewards.

## B.2. Regret Bounds of DAL in Piecewise Stationary Environments

As discussed in Section 4.2 of the paper, using Corollary 4.9, we can select different algorithms as input for DAL to attain or improve the state-of-the-art regret bounds in PS environments. Combining DAL with different bandit algorithms leads to the results in Table 1. It is evident that DAL matches the state-of-the-art regret bounds in PS-LBs, PS-GLBs and PS-CBs, and DAL improves the best known bounds in the PS-SCB and PS-KB settings. Note that for PS-SCBs, the strongest result corresponds to the prior-based WeightUCB (Wang et al., 2023). As demonstrated in the final columns of the table, the order-wise dependence on problem parameters from the stationary setting seamlessly transfers to the PS setting without degradation.

## B.3. Proof of Proposition 4.2

In the NS-PB setting, the reward at time $t$ is given by $f_t(a) = \mu(\langle \theta_t, a \rangle)$ for all $a \in \mathcal{A}$, where $\mu$ is injective and $\theta_t \in \mathbb{R}^d$. To detect any changes in $\theta_t$, it suffices to detect changes in the values $\langle \theta_t, a \rangle$ for a suitable set of actions.

Since $\mu$ is injective, each observation $y_{t,i} = \mu(\langle \theta_t, a_i \rangle)$ can be inverted to recover the inner product:

$$\langle \theta_t, a_i \rangle = \mu^{-1}(y_{t,i}).$$

*Table 2.* Regret bound comparison of algorithms for PS bandits, under the Assumption 4.6. "†" denotes settings with finite number of actions, while MASTER, ADA-OPKB and SCB-WeightUCB also recover the appropriate bounds in this setting. "●" indicates prior-based algorithms.

| PS Setting | Non-Stationary Algorithm | NS Algorithm Regret Bound in $\tilde{\mathcal{O}}(\cdot)$ | DAL Input Regret Bound in $\tilde{\mathcal{O}}(\cdot)$ |
|---|---|---|---|
| PS-LB | MASTER (Wei & Luo, 2021) + LinUCB | $d\sqrt{TN_T}$ | - |
| | ADA-OPKB (Hong et al., 2023) | $d\sqrt{N_T T}$ | - |
| | DAL (ours) + LinUCB (Abbasi-yadkori et al., 2011) | $d\sqrt{N_T T}$ | $d\sqrt{T}$ |
| | DAL (ours) + LinTS (Agrawal & Goyal, 2013) | $d^{3/2}\sqrt{N_T T}$ | $d^{3/2}\sqrt{T}$ |
| | DAL (ours) + PEGE$^\dagger$ (Lattimore & Szepesvári, 2020) | $\sqrt{dN_T T}$ | $\sqrt{dT}$ |
| PS-GLB | MASTER (Wei & Luo, 2021) + GLM-UCB | $d\sqrt{N_T T}$ | - |
| | DAL (ours) + GLM-UCB (Filippi et al., 2010) | $d\sqrt{N_T T}$ | $d\sqrt{T}$ |
| | DAL (ours) + GLM-TSL (Kveton et al., 2020) | $d^{3/2}\sqrt{N_T T}$ | $d^{3/2}\sqrt{T}$ |
| | DAL (ours) + SupCB-GLM$^\dagger$ (Li et al., 2017) | $\sqrt{dN_T T}$ | $\sqrt{dT}$ |
| PS-SCB | SCB-WeightUCB$^\bullet$ (Wang et al., 2023) | $d^{2/3}T^{2/3}N_T^{1/3}$ | – |
| | DAL (ours) + OFU-ECOLog (Faury et al., 2022) | $d\sqrt{N_T T}$ | $d\sqrt{T}$ |
| | DAL (ours) + OFUL-MLogB (Zhang & Sugiyama, 2023) | $d\sqrt{N_T T}$ | $d\sqrt{T}$ |
| | DAL (ours) + OFUGLB (Lee et al., 2024) | $d\sqrt{N_T T}$ | $d\sqrt{T}$ |
| PS-KB | MASTER (Wei & Luo, 2021) + GPUCB | $\gamma_T\sqrt{N_T T}$ | - |
| | ADA-OPKB (Hong et al., 2023) | $\sqrt{d\gamma_T N_T T}$ | - |
| | DAL (ours) + GPUCB (Chowdhury & Gopalan, 2017) | $\gamma_T\sqrt{N_T T}$ | $\gamma_T\sqrt{T}$ |
| | DAL (ours) + REDS (Salgia et al., 2024) | $\sqrt{\gamma_T N_T T}$ | $\sqrt{\gamma_T T}$ |
| PS-CB | MASTER (Wei & Luo, 2021) + ILTCB | $\sqrt{|\mathcal{A}|N_T T \log|\Pi|}$ | - |
| | ADA-ILTCB+ (Chen et al., 2019) | $\sqrt{|\mathcal{A}|N_T T \log|\Pi|}$ | - |
| | DAL (ours) + ILTCB (Agarwal et al., 2014) | $\sqrt{|\mathcal{A}|N_T T \log|\Pi|}$ | $\sqrt{|\mathcal{A}|T \log|\Pi|}$ |
| | DAL (ours) + SquareCB (Foster & Rakhlin, 2020) | $\sqrt{|\mathcal{A}|N_T T \log|\Pi|}$ | $\sqrt{|\mathcal{A}|T \log|\Pi|}$ |

Hence, observing $y_{t,i}$ is equivalent to observing $\langle \theta_t, a_i \rangle$.

Suppose that $\mathcal{A}_e \subseteq \mathcal{A}$ is the maximal linearly independent subset of $\mathcal{A}$. Then, the vector $\theta_t$ is uniquely determined by the inner products $\langle \theta_t, a \rangle$ for $a \in \mathcal{A}_e$. Therefore, any change in $\theta_t$ results in a detectable change in the vector of observations $(y_{t,i})_{a_i \in \mathcal{A}_e}$, meaning that $\mathcal{A}_e$ can be taken to be any maximal linearly independent subset of $\mathcal{A}$, with $|\mathcal{A}_e| \leq d$.

### B.4. Proof of Proposition 4.3

In this subsection, we establish the construction of $\mathcal{A}_e$ in the NS-KB setting. According to Lemma 5 from (De Freitas et al., 2012), we have that every $f \in H_k$ with $\|f\|_{H_k} \leq B$ is Lipschitz continuous, satisfying the following,

$$|f(x) - f(y)| \leq B L_u \|x - y\|_2, \ \forall \, x, y \in \mathcal{A}, \quad \text{where } L_u := \sup_{z \in D} \max_{i,j \leq d} \Big[ \frac{\partial^2 k(p, q)}{\partial p_i \, \partial q_j} \Big]^{1/2}_{p=q=z}.$$

Recall that $\mathcal{V}_T$ corresponds to the set of centers of the balls of an arbitrary $\delta_T$-cover of $\mathcal{A} \subseteq [0, R]^d$, with $\delta_T = L_T/(2BL_u)$ for some arbitrary $L_T > 0$. Let $[a]_e$ denote the action in $\mathcal{V}_T$ that is the closest to $a \in \mathcal{A}$, i.e., $[a]_e = \operatorname{argmin}_{x \in \mathcal{P}_T} \|a - x\|_2$. Then, we can leverage the Lipschitz property of functions in the RKHS to obtain the following upper bound: For any $a \in \mathcal{A}$ and $f \in H_k$ with $\|f\|_{H_k} \leq B$,

$$|f(a) - f([a]_e)| \overset{(a)}{\leq} BL_u\|a - [a]_e\|_2 \overset{(b)}{\leq} BL_u\delta_T. \tag{1}$$

Step $(a)$ follows from the Lipschitz property in Lemma 5 of (De Freitas et al., 2012), and step $(b)$ results from the definition of a $\delta_T$-cover. Then, for any arbitrary functions $f$ and $f'$ in $H_k$ with $\|f\|_{H_k}, \|f'\|_{H_k} \leq B$ and action $\tilde{a} \in \mathcal{A}$, we have

$$|f([\tilde{a}]_e) - f'([\tilde{a}]_e)| \geq |f(\tilde{a}) - f'(\tilde{a})| - |f(\tilde{a}) - f([\tilde{a}]_e)| - |f'(\tilde{a}) - f'([\tilde{a}]_e)|$$

$$\overset{(a)}{\geq} |f(\tilde{a}) - f'(\tilde{a})| - 2BL_u\delta_T = |f(\tilde{a}) - f'(\tilde{a})| - L_T \overset{(b)}{>} 0$$

where step $(a)$ is due to (1), and step $(b)$ is due to the assumption in Proposition 4.3. This indicates that the value of the reward function at $[\tilde{a}]_e$ must change by a non-zero amount. Thus, one can use observations from action $[\tilde{a}]_e$ in order to deduce whether the reward function has changed its value in action $\tilde{a}$. In addition, by the upper bound on the covering number, the cardinality of $\mathcal{V}_T$ is upper bounded by $\lceil \sqrt{d}R/2\delta_T \rceil^d = \lceil \sqrt{d}BL_uR/L_T \rceil^d$.

### B.5. Proof of Theorem 4.8

For PS-PBs and PS-KBs, the proof of Theorem 4.8 follows exactly the same as those of Theorem 1 and Corollary 1 in (Huang et al., 2025), with the number of arms replaced by $N_e$, due to the different number of actions in the covering set. For completeness, we provide a proof sketch of Theorem 4.8: First, we partition the regret into two cases. If no false alarm occurs and all changes are detected within a short delay, we can separate the regret into three components: the regret due to forced exploration, the regret during the short detection (restart) delay after changes, and the regret incurred by the stationary bandit algorithm after the change is detected. If not, we use a crude linear bound to bound the regret and show that the probability of false alarm and that of late detection are low, which ensures that the regret due to detection failure is small.

For PS-CBs, the proof of Theorem 4.8 the definition of successful detection events should be modified as follows:

Consider a PS-CB environment satisfying the change-point separation condition in Theorem 4.8, and recall that $\mathcal{D}$ is the change detector of DAL. Let $\tau_k$ be the $k^{\text{th}}$ detection point for $k \in \mathbb{N}$, i.e.,

$$\tau_k := \inf \{t > \tau_{k-1} : \mathcal{D}(H_{c,a}) = \text{Detection at time-step } t \text{ for some } (c,a) \in \mathcal{C} \times \mathcal{A}_e\}, \tag{2}$$

where $\tau_0 = 0$. Recall that $\nu_0 := 1$ and $\nu_{N_T+1} := T + 1$. We define the following events:

$$\mathcal{G}_k := \{\forall l \in [k-1], \ \tau_l \in \{\nu_l, \ldots, \nu_l + \ell_l - 1\}\} \cap \{\tau_k > \nu_k\}, \ k \in [N_T]. \tag{3}$$

The event $\mathcal{G}_k$ represents the "good event" up to the $k^{\text{th}}$ detection point $\mathcal{G}_k$ in which the first $k$ changes are detected within the latency. For notational convenience, we define $\mathcal{G}_0$ to be the universal space. Then, we have the following:

$$
\begin{aligned}
R_T &= \mathbb{E}\left[\sum_{k=1}^{N_T+1} \sum_{t=\nu_{k-1}}^{\nu_k-1} \max_{\pi \in \Pi} f_t(C_t, \pi(C_t)) - f_t(C_t, A_t)\right] \\
&= \sum_{k=1}^{N_T+1} \mathbb{E}\left[\sum_{t=\nu_{k-1}}^{\nu_k-1} \max_{\pi \in \Pi} f_t(C_t, \pi(C_t)) - f_t(C_t, A_t)\right] \\
&= \sum_{k=1}^{N_T+1} \mathbb{P}(\mathcal{G}_k^c) \mathbb{E}\left[\sum_{t=\nu_{k-1}}^{\nu_k-1} \max_{\pi \in \Pi} f_t(C_t, \pi(C_t)) - f_t(C_t, A_t)\Big| \mathcal{G}_k^c\right] \\
&\quad + \sum_{k=1}^{N_T+1} \mathbb{E}\left[\mathbb{1}\{\mathcal{G}_k\} \sum_{t=\nu_{k-1}}^{\nu_k-1} \max_{\pi \in \Pi} f_t(C_t, \pi(C_t)) - f_t(C_t, A_t)\right] \\
&\stackrel{(a)}{\leq} \sum_{k=1}^{N_T+1} \bar{\Delta}(\nu_k - \nu_{k-1}) \mathbb{P}(\mathcal{G}_k^c) + \sum_{k=1}^{N_T+1} \mathbb{E}\left[\mathbb{1}\{\mathcal{G}_k\} \sum_{t=\nu_{k-1}}^{\nu_k-1} \max_{\pi \in \Pi} f_t(C_t, \pi(C_t)) - f_t(C_t, A_t)\right] \tag{4}
\end{aligned}
$$

where $\bar{\Delta}$ in step $(a)$ is the maximum gap between the mean rewards of two actions, over all contexts, actions, and time-steps, i.e., $\bar{\Delta} := \max_{c \in \mathcal{C}, a \in \mathcal{A}, t \in [T]} (\max_{\pi \in \Pi} f_t(c, \pi(c)) - f_t(c, a))$. For convenience in the proof of the upper bound on the probability of bad event $\mathbb{P}(\mathcal{G}_k^c)$, define

$$\mathcal{E}_k := \{\forall l \in [k-1], \ \tau_l \in \{\nu_l, \ldots, \nu_l + \ell_l - 1\}\}, \ k \in [N_T]. \tag{5}$$

$\mathbb{P}(\mathcal{G}_k^c)$ is upper bounded by the following modified union bound, which decomposes the bad event into false alarm events

and late detection events:

$$
\begin{aligned}
\mathbb{P}\left(\mathcal{G}_k^c\right) &= \mathbb{P}\left(\left\{\exists\, l \in [k-1],\ \tau_l \notin \{\nu_l, \ldots, \nu_l + \ell_l - 1\}\right\} \cup \{\tau_k \leq \nu_k\}\right) \\
&= \sum_{l=1}^{k-1} \mathbb{P}\left(\tau_l \notin \{\nu_s, \ldots, \nu_l + \ell_l - 1\}, \mathcal{E}_{l-1}\right) + \mathbb{P}\left(\tau_k \leq \nu_k, \mathcal{E}_{k-1}\right) \\
&= \sum_{l=1}^{k-1} \mathbb{P}\left(\mathcal{E}_{l-1}\right) \mathbb{P}\left(\tau_l \notin \{\nu_l, \ldots, \nu_l + \ell_l - 1\} \,\middle|\, \mathcal{E}_{l-1}\right) + \mathbb{P}\left(\mathcal{E}_{k-1}\right) \mathbb{P}\left(\tau_k \leq \nu_k \,\middle|\, \mathcal{E}_{k-1}\right) \\
&\overset{(a)}{\leq} \sum_{l=1}^{k-1} \mathbb{P}\left(\tau_l \notin \{\nu_l, \ldots, \nu_l + \ell_l - 1\} \,\middle|\, \mathcal{E}_{l-1}\right) + \mathbb{P}\left(\tau_k \leq \nu_k \,\middle|\, \mathcal{E}_{k-1}\right) \\
&= \sum_{l=1}^{k} \underbrace{\mathbb{P}\left(\tau_l < \nu_l \,\middle|\, \mathcal{E}_{l-1}\right)}_{\Phi_1} + \sum_{l=1}^{k-1} \underbrace{\mathbb{P}\left(\tau_l \geq \nu_l + \ell_l \,\middle|\, \mathcal{E}_{l-1}\right)}_{\Phi_2}
\end{aligned}
\tag{6}
$$

where $(a)$ is due to the fact that $\mathbb{P}\{\mathcal{E}_{k-1}\} \leq 1$. We then separately bound $\Phi_1$ and $\Phi_2$.

• *Upper-Bounding* $\Phi_1$: Recall that $\mathcal{A}_e = \left\{a^{(i)}, i \in [N_e]\right\}$ is the covering set, and that $H_{(c, a^{(i)})}$ is the change detector history list associated with the context-action pair $\left(c, a^{(i)}\right)$. For any context $c \in \mathcal{C}$, $i \in [N_e]$, and $u \in \mathbb{N}$, we define $t'_{(c,i),u}$ to be the $u^{\text{th}}$ time-step after $\tau_{l-1}$ at which $C_t = c$ and $(t - \tau_{l-1} - 1) \mod \lceil N_e/\alpha_l \rceil = i - 1$, i.e.,

$$
t'_{(c,i),u} := \inf\left\{t > t'_{(c,i),u-1} : C_t = c, (t - \tau_{l-1} - 1) \mod \left\lceil \frac{N_e}{\alpha_l} \right\rceil = i - 1\right\}
\tag{7}
$$

where $t'_{(c,i),0} = \tau_{l-1}$. Then, we define $n_{c,i}(t)$ to be the number of time-steps between $\tau_{l-1} + 1$ and $t$ at which $C_t = c$ and $(t - \tau_{l-1} - 1) \mod \lceil N_e/\alpha_l \rceil = i - 1$, which is the number of samples obtained due to force exploration and added in the history $H_{(c, a^{(i)})}$ if there are no restarts after $\tau_{l-1}$, i.e.,

$$
n_{(c,i)}(t) := \sum_{s = \tau_{l-1}+1}^{t} \mathbb{1}\left\{C_t = c, (t - \tau_{l-1} - 1) \mod \left\lceil \frac{N_e}{\alpha_l} \right\rceil = i - 1\right\}.
\tag{8}
$$

We also use $\tau_{(c,i)}$ to denote the stopping time of the change detector associated with arm $a^{(i)} \in \mathcal{A}_e$ after the $(l-1)^{\text{th}}$ detection point $\tau_{l-1}$, i.e.,

$$
\tau_{(c,i)} := \inf\left\{u \in \mathbb{N} : \mathcal{D}\left(H_{c, a^{(i)}}\right) = \text{Detection at time-step } t'_{(c,i),u}\right\}.
\tag{9}
$$

The stopping time $\tau_{(c,i)}$ operates independently from other stopping times, and does not stop if other stopping times get triggered earlier. Let $\mathbb{P}_\infty$ denote the probability measure at which $f_t = f_{\nu_l}$ for all $t > \nu_l$, i.e., the probability measure under which the CB becomes stationary after the $k^{\text{th}}$ change-point. Then, for all $l \in [N_T + 1]$, we have

$$
\begin{aligned}
\mathbb{P}\left(\tau_l < \nu_l \,\middle|\, \mathcal{E}_{l-1}\right) &= \mathbb{P}\left(\exists\, \left(c, a^{(i)}\right) \in \mathcal{C} \times \mathcal{A}_e : \tau_{(c,i)} \in \left[n_{(c,i)}(\nu_l - 1)\right] \,\middle|\, \mathcal{E}_{l-1}\right) \\
&\overset{(a)}{\leq} \sum_{c \in \mathcal{C}} \sum_{i=1}^{N_e} \mathbb{P}\left(\tau_{(c,i)} \in \left[n_{(c,i)}(\nu_l - 1)\right] \,\middle|\, \mathcal{E}_{l-1}\right) \\
&\overset{(b)}{\leq} \sum_{c \in \mathcal{C}} \sum_{i=1}^{N_e} \mathbb{P}\left(\tau_{(c,i)} \leq T \,\middle|\, \mathcal{E}_{l-1}\right) \\
&\overset{(c)}{\leq} \sum_{c \in \mathcal{C}} \sum_{i=1}^{N_e} \delta_F \\
&= |\mathcal{C}| N_e \delta_F
\end{aligned}
\tag{10}
$$

where step $(a)$ results from a union bound. Due to the fact that the rewards between $\tau_{l-1}$ and $\nu_l$ are i.i.d. across time-steps and actions given the past event $\mathcal{E}_{l-1}$ (as there are no changes between $\tau_{l-1}$ and $\nu_l$), we can change the measure to $\mathbb{P}_\infty$

in step $(b)$. In addition, because $[n_a\,(\nu_l - 1)] \subseteq [T]$, the event $\{\tau_{a,l} \in [n_a\,(\nu_l - 1)]\} \subseteq \{\tau_{a,l} \le T\}$. In step $(c)$, since the reward samples $\{X_{t'_{(c,i),u}}\}_{u \ge 1}$ are i.i.d. sub-Gaussian for each $(c, i) \in \mathcal{C} \times [N_e]$, we can apply the false alarm probability upper bound for the GLR and GSR tests in (Huang & Veeravalli, 2025) (see Section 4.1).

- *Upper Bounding* $\Phi_2$: Let $(c^*, i^*)$ be the context-action pair at which the mean reward function changes the most at $\nu_l$, i.e.,

$$(c^*, i^*) = \underset{c \in \mathcal{C}, i \in [N_e]}{\operatorname{argmax}} \left| f_{\nu_l}\left(c, a^{(i)}\right) - f_{\nu_l - 1}\left(c, a^{(i)}\right) \right|. \tag{11}$$

We define the events $\mathcal{M}_l$ and $\mathcal{L}_l$ as follows:

$$\mathcal{M}_l := \left\{ \sum_{t=\tau_{l-1}+1}^{\nu_l - 1} \mathbb{1}\left\{ C_t = c^*, (t - \tau_{l-1} - 1) \bmod \left\lceil \frac{N_e}{\alpha_l} \right\rceil = i^* - 1 \right\} \ge m_{\mathcal{D}} \right\}, \tag{12}$$

$$\mathcal{L}_l := \left\{ \sum_{t=\nu_l}^{\nu_l + \ell_l - 1} \mathbb{1}\left\{ C_t = c^*, (t - \tau_{l-1} - 1) \bmod \left\lceil \frac{N_e}{\alpha_l} \right\rceil = i^* - 1 \right\} \ge \ell_{\mathcal{D}} \right\}. \tag{13}$$

When $\tau_l \ge \nu_l + \ell_l$, there are at least $m_{\mathcal{D}}$ reward samples following $f_{\nu_l - 1}$ in $H_{(c^*, a^{(i^*)})}$ under the event $\mathcal{M}_l$, and there are at least $\ell_{\mathcal{D}}$ reward samples following $f_{\nu_l}$ in $H_{(c^*, a^{(i^*)})}$ under the event $\mathcal{L}_l$. Then, we have,

$$\begin{aligned}
&\mathbb{P}\left(\tau_l \ge \nu_l + \ell_l \middle| \mathcal{E}_{l-1}\right) \\
&\le \mathbb{P}\left(\{\tau_l \ge \nu_l + \ell_l\} \cup \mathcal{M}_l^c \cup \mathcal{L}_l^c \middle| \mathcal{E}_{l-1}\right) \\
&= \mathbb{P}\left(\mathcal{M}_l^c \cup \mathcal{L}_l^c \middle| \mathcal{E}_{l-1}\right) + \mathbb{P}\left(\{\tau_l \ge \nu_l + \ell_l\} \cap \mathcal{M}_l \cap \mathcal{L}_l \middle| \mathcal{E}_{l-1}\right) \\
&= \mathbb{P}\left(\mathcal{M}_l^c \cup \mathcal{L}_l^c \middle| \mathcal{E}_{l-1}\right) + \mathbb{P}\left(\mathcal{M}_l \cap \mathcal{L}_l \middle| \mathcal{E}_{l-1}\right) \mathbb{P}\left(\tau_l \ge \nu_l + \ell_l \middle| \mathcal{M}_l \cap \mathcal{L}_l \cap \mathcal{E}_{l-1}\right) \\
&\overset{(a)}{\le} \mathbb{P}\left(\mathcal{M}_l^c \middle| \mathcal{E}_{l-1}\right) + \mathbb{P}\left(\mathcal{L}_l^c \middle| \mathcal{E}_{l-1}\right) + \mathbb{P}\left(\tau_l \ge \nu_l + \ell_l \middle| \mathcal{M}_l \cap \mathcal{L}_l \cap \mathcal{E}_{l-1}\right)
\end{aligned} \tag{14}$$

where step $(a)$ follows from a union bound and the fact that $\mathbb{P}\left(\mathcal{M}_l \cap \mathcal{L}_l \middle| \mathcal{E}_{l-1}\right) \le 1$. For upper bounding the first two terms, we use the fact that given $\mathcal{E}_{l-1}$, for any $i \in [N_e]$ and $u, v > \tau_{l-1}$ where $v < u$,

$$\sum_{t=v+1}^{u} \mathbb{1}\left\{ (t - \tau_k - 1) \bmod \left\lceil \frac{N_e}{\alpha_l} \right\rceil = i - 1 \right\} \ge \left\lfloor \frac{u - v}{\lceil N_e/\alpha_l \rceil} \right\rfloor. \tag{15}$$

The inequality in (15) holds with equality when $u - v$ is divisible by $\lceil N_e/\alpha_l \rceil$. Recall that $n_{(c,i)}(t)$ is the number of time-steps between $\tau_{l-1} + 1$ and $t$ at which $C_t = c$ and $(t - \tau_{l-1} - 1 \bmod \lceil N_e/\alpha_l \rceil) = i - 1$ (see (8)). Then, we have

$$\begin{aligned}
&\mathbb{E}\left[n_{(c^*,i^*)}(\nu_l - 1) - n_{(c^*,i^*)}(\tau_{l-1}) \middle| \mathcal{E}_{l-1}\right] \\
&\overset{(a)}{\ge} \mathbb{E}\left[n_{(c^*,i^*)}(\nu_l - 1) - n_{(c^*,i^*)}(\nu_l - m_l - 1) \middle| \mathcal{E}_{l-1}\right] \\
&= \mathbb{E}\left[ \sum_{t=\nu_l - m_l}^{\nu_l - 1} \mathbb{1}\left\{ C_t = c^*, (t - \tau_l - 1) \bmod \left\lceil \frac{N_e}{\alpha_l} \right\rceil = i^* - 1 \right\} \middle| \mathcal{E}_{l-1}\right] \\
&= \sum_{t=\nu_l - m_l}^{\nu_l - 1} \mathbb{P}\left(C_t = c^* \middle| \mathcal{E}_{l-1}\right) \mathbb{1}\left\{ (t - \tau_l - 1) \bmod \left\lceil \frac{N_e}{\alpha_l} \right\rceil = i^* - 1 \right\} \\
&\overset{(b)}{=} \sum_{t=\nu_l - m_l}^{\nu_l - 1} \mathcal{P}_t(c) \mathbb{1}\left\{ (t - \tau_l - 1) \bmod \left\lceil \frac{N_e}{\alpha_l} \right\rceil = i^* - 1 \right\} \\
&\overset{(c)}{\ge} s \sum_{t=\nu_l - m_l}^{\nu_l - 1} \mathbb{1}\left\{ (t - \tau_l - 1) \bmod \left\lceil \frac{N_e}{\alpha_l} \right\rceil = i^* - 1 \right\} \\
&\overset{(d)}{=} s \left\lfloor \frac{m_l}{\lceil N_e/\alpha_l \rceil} \right\rfloor \\
&= s \left\lceil \frac{m_{\mathcal{D}}}{s} + \frac{\log T}{4s^2} + \sqrt{\frac{m_{\mathcal{D}} \log T}{2s^3} + \frac{(\log T)^2}{16s^4}} \right\rceil,
\end{aligned} \tag{16}$$

and

$$\mathbb{E}\left[n_{(c^*,i^*)}\left(\nu_l + \ell_l - 1\right) - n_{(c^*,i^*)}\left(\nu_l - 1\right)\right]$$

$$= \mathbb{E}\left[\sum_{t=\nu_l}^{\nu_l+\ell_l-1} \mathbb{1}\left\{C_t = c^*, (t - \tau_l - 1) \mod \left\lceil \frac{N_{\mathrm{e}}}{\alpha_l} \right\rceil = i^* - 1\right\}\right]$$

$$= \sum_{t=\nu_l}^{\nu_l+\ell_l-1} \mathbb{P}\left(C_t = c^* | \mathcal{E}_{l-1}\right) \mathbb{1}\left\{(t - \tau_l - 1) \mod \left\lceil \frac{N_{\mathrm{e}}}{\alpha_l} \right\rceil = i^* - 1\right\}$$

$$\overset{(e)}{=} \sum_{t=\nu_l}^{\nu_l+\ell_l-1} \mathcal{P}_t(c) \mathbb{1}\left\{(t - \tau_l - 1) \mod \left\lceil \frac{N_{\mathrm{e}}}{\alpha_l} \right\rceil = i^* - 1\right\}$$

$$\overset{(f)}{\geq} s \sum_{t=\nu_l}^{\nu_l+\ell_l-1} \mathbb{1}\left\{(t - \tau_l - 1) \mod \left\lceil \frac{N_{\mathrm{e}}}{\alpha_l} \right\rceil = i^* - 1\right\}$$

$$\overset{(g)}{=} s \left\lfloor \frac{\ell_l}{\lceil N_{\mathrm{e}}/\alpha_l \rceil} \right\rfloor$$

$$= s \left\lceil \frac{\ell_{\mathcal{D}}}{s} + \frac{\log T}{4s^2} + \sqrt{\frac{\ell_{\mathcal{D}} \log T}{2s^3} + \frac{(\log T)^2}{16s^4}} \right\rceil. \tag{17}$$

In step $(a)$, since $\tau_{l-1} \leq \nu_{l-1} + \ell_{l-1} - 1$ given $\mathcal{E}_{l-1}$ and $\nu_l - \nu_{l-1} \geq \ell_{l-1} + m_l$ by Assumption 4.6, $\tau_{l-1} \leq \nu_l - m_l - 1$ and thus $n_{(c^*,i^*)}(\nu_l - 1) \leq n_{(c^*,i^*)}(\nu_l - m_l - 1)$. Steps $(b)$ and $(e)$ follow from the independence between $(C_t)_{t>\tau_l}$ and $\mathcal{E}_{l-1}$. Steps $(c)$ and $(f)$ stem from the definition of $s$ in Theorem 4.6 ($s = \min_{c \in \mathcal{C}, t \in [T]: \mathcal{P}_t(c) > 0} \mathcal{P}_t(c)$). Steps $(d)$ and $(g)$ result from (15), as $m_l$ and $\ell_l$ are divisible by $\lceil N_{\mathrm{e}}/\alpha_l \rceil$. Therefore,

$$\mathbb{P}\left(\mathcal{M}_l^c | \mathcal{E}_{l-1}\right)$$

$$= \mathbb{P}\left(\sum_{t=\tau_l+1:(t-\tau_k-1) \mod \lceil N_{\mathrm{e}}/\alpha_l \rceil = i^*-1}^{\nu_l-1} \mathbb{1}\left\{C_t = c^*\right\} \leq m_{\mathcal{D}} \Big| \mathcal{E}_{l-1}\right)$$

$$\overset{(a)}{\leq} \exp\left(\frac{-2\left(\mathbb{E}\left[n_{(c^*,i^*)}(\nu_l-1) - n_{(c^*,i^*)}(\tau_{l-1})\right] - m_{\mathcal{D}}\right)^2}{\sum_{t=\tau_l+1}^{\nu_l-1} \mathbb{1}\left\{(t-\tau_l-1) \mod \lceil N_{\mathrm{e}}/\alpha_l \rceil = i^*-1\right\}}\right)$$

$$\overset{(b)}{\leq} \exp\left(\frac{-2\left(s\left\lceil m_{\mathcal{D}}/s + \log(T)/4s^2 + \sqrt{m_{\mathcal{D}} \log T/2s^3 + (\log T)^2/16s^4}\right\rceil - m_{\mathcal{D}}\right)^2}{\left\lceil m_{\mathcal{D}}/s + \log(T)/4s^2 + \sqrt{m_{\mathcal{D}} \log T/2s^3 + (\log T)^2/16s^4}\right\rceil}\right)$$

$$\leq T^{-1}, \tag{18}$$

and

$$\mathbb{P}\left(\mathcal{L}_l^c | \mathcal{E}_{l-1}\right)$$

$$= \mathbb{P}\left(\sum_{t=\nu_l:(t-\tau_k-1) \mod \lceil N_{\mathrm{e}}/\alpha_l \rceil = i^*-1}^{\nu_l+\ell_l-1} \mathbb{1}\left\{C_t = c^*\right\} \leq \ell_{\mathcal{D}} \Big| \mathcal{E}_{l-1}\right)$$

$$\overset{(c)}{\leq} \exp\left(\frac{-2\left(\mathbb{E}\left[n_{(c^*,i^*)}(\nu_l+\ell_l-1) - n_{(c^*,i^*)}(\nu_l-1)\right] - \ell_{\mathcal{D}}\right)^2}{\sum_{t=\nu_l}^{\nu_l+\ell_l-1} \mathbb{1}\left\{(t-\tau_l-1) \mod \lceil N_{\mathrm{e}}/\alpha_l \rceil = i^*-1\right\}}\right)$$

$$\overset{(d)}{\leq} \exp\left(\frac{-2\left(s\left\lceil \ell_{\mathcal{D}}/s + \log(T)/4s^2 + \sqrt{\ell_{\mathcal{D}} \log T/2s^3 + (\log T)^2/16s^4}\right\rceil - \ell_{\mathcal{D}}\right)^2}{\left\lceil \ell_{\mathcal{D}}/s + \log(T)/4s^2 + \sqrt{\ell_{\mathcal{D}} \log T/2s^3 + (\log T)^2/16s^4}\right\rceil}\right)$$

$$\leq T^{-1}. \tag{19}$$

In steps $(a)$ and $(c)$, we apply Hoeffding's inequality, as $\{\mathbb{1}\{C_t = c^*\}\}_{t \geq \tau_l}$ is a sequence of i.i.d. Bernoulli random variables with parameter $\mathcal{P}_t(c)$. In steps $(b)$ and $(d)$, we apply (17).

Before bounding the third term in (14), recall the definition of the stopping time of the change detector associated with arm $a^{(i)}$ after the $(l-1)^{\text{th}}$ detection point in (9). Without loss of generality, we assume that $\nu_l \leq T - \ell_l$; otherwise, there is no need to detect the change because the horizon will end soon after the change occurs. We can derive that

$$
\begin{aligned}
&\mathbb{P}\left(\tau_l \geq \nu_l + \ell_l \big| \mathcal{E}_{l-1} \cap \mathcal{M}_l \cap \mathcal{L}_l\right) \\
&= \mathbb{P}\left(\forall (c,i) \in \mathcal{C} \times [N_{\mathrm{e}}],\ \tau_{(c,i)} > n_{(c,i)}\left(\nu_l + \ell_l - 1\right) \big| \mathcal{E}_{l-1} \cap \mathcal{M}_l \cap \mathcal{L}_l\right) \\
&\overset{(a)}{\leq} \mathbb{P}\left(\tau_{(c^*,i^*)} > n_{(c^*,i^*)}\left(\nu_l + \ell_l - 1\right) \big| \mathcal{E}_{l-1} \cap \mathcal{M}_l \cap \mathcal{L}_l\right) \\
&\overset{(b)}{\leq} \mathbb{P}\left(\tau_{(c^*,i^*)} > n_{(c^*,i^*)}\left(\nu_l - 1\right) + \ell_{\mathcal{D}} \big| \mathcal{E}_{l-1} \cap \mathcal{M}_l \cap \mathcal{L}_l\right) \\
&\overset{(c)}{\leq} \sup_{\nu \in \{m_{\mathcal{D}}+1, \ldots, T-\ell_{\mathcal{D}}\}} \mathbb{P}\left(\tau_{(c^*,i^*)} \geq \nu + \ell_{\mathcal{D}} \big| \mathcal{E}_{l-1} \cap \mathcal{M}_l \cap \mathcal{L}_l\right) \\
&\overset{(d)}{\leq} \delta_{\mathrm{D}}
\end{aligned}
\tag{20}
$$

where step $(a)$ comes from the fact that $\{(c^*, i^*)\} \subseteq \mathcal{C} \times [N_{\mathrm{e}}]$, and step $(b)$ stems from the fact that $n_{(c^*,i^*)}\left(\nu_l + \ell_l - 1\right) - n_{(c^*,i^*)}\left(\nu_l - 1\right) \geq \ell_{\mathcal{D}}$ given $\mathcal{L}_l$. Step $(c)$ results from the fact that $n_{(c^*,i^*)}\left(\nu_l - 1\right) \geq m_{\mathcal{D}}$ given $\mathcal{M}_l$ and $\nu_l \leq T - \ell_l$. Recall the definition of $t'_{(c,i),u}$ in (7). Step $(d)$ follows from the definition of latency in Section 4.1, as the reward sequence $\left\{X_{t'_{(c^*,i^*),u}}\right\}_{u \geq 1}$ are independent sub-Gaussian whose distribution changes at $\nu$, given $\mathcal{E}_{l-1}$ and the context sequence $\{C_t\}_{t \geq 1}$. Plugging (18), (19), and (20) into (14), we have

$$
\mathbb{P}\left(\tau_l \geq \nu_l + \ell_l \big| \mathcal{E}_{l-1}\right) \leq 2T^{-1} + \delta_{\mathrm{D}}.
\tag{21}
$$

This completes bounding $\Phi_1$ and $\Phi_2$. Plugging (10) and (20) into (6), we obtain

$$
\mathbb{P}\{\mathcal{G}_k^c\} \leq |\mathcal{C}| N_{\mathrm{e}} k \delta_{\mathrm{F}} + (k-1)\left(2T^{-1} + \delta_{\mathrm{D}}\right).
\tag{22}
$$

This bounds the first term in (4).

For convenience in bounding the second term in (4), we define $\bar{\alpha} := \max_{k=1,\ldots,N_T+1} \alpha_k$. Recall that $\bar{\Delta} = \max_{c \in \mathcal{C}, a \in \mathcal{A}, t \in [T]}\left(\max_{\pi \in \Pi} f_t(c, \pi(c)) - f_t(c, a)\right)$. For any $k \in [N_T + 1]$, if $(t - \tau_{k-1} - 1 \mod \lceil N_{\mathrm{e}}/\alpha_k \rceil) \geq N_{\mathrm{e}}$, then $A_t$ follows the stationary CB algorithm $\mathcal{B}$. Thus, the second term in (4) can then be decomposed as follows:

$$
\begin{aligned}
&\mathbb{E}\left[\mathbb{1}\{\mathcal{G}_k\} \sum_{t=\nu_{k-1}}^{\nu_k - 1} \max_{\pi \in \Pi} f_t(C_t, \pi(C_t)) - f_t(C_t, A_t)\right] \\
&\overset{(a)}{\leq} \bar{\Delta}\ell_{k-1} + \bar{\Delta} N_{\mathrm{e}} \left\lceil \frac{\nu_k - \nu_{k-1}}{\lceil N_{\mathrm{e}}/\alpha_k \rceil} \right\rceil \\
&\quad + \mathbb{E}\left[\mathbb{1}\{\mathcal{G}_k\} \sum_{t=\tau_{k-1}+1:(t-\tau_{k-1}-1) \mod \lceil N_{\mathrm{e}}/\alpha_k \rceil \geq N_{\mathrm{e}}}^{\nu_k-1} \left(\max_{\pi \in \Pi} f_t(C_t, \pi(C_t)) - f_t(C_t, A_t)\right)\right] \\
&\overset{(b)}{\leq} \bar{\Delta}\ell_{k-1} + \bar{\Delta}\left[\alpha_k(\nu_k - \nu_{k-1}) + N_{\mathrm{e}}\right] + R_{\mathcal{B}}(\nu_k - \nu_{k-1}) \\
&\leq \bar{\Delta}\ell_{k-1} + \bar{\Delta}\left[\bar{\alpha}(\nu_k - \nu_{k-1}) + N_{\mathrm{e}}\right] + R_{\mathcal{B}}(\nu_k - \nu_{k-1})
\end{aligned}
\tag{23}
$$

where in step $(a)$, the first term bounds the regret due to the delay of the change detector, and the second term bounds the regret incurred due to forced exploration. In step $(b)$, as the reward samples in the history of the stationary bandit algorithm $\mathcal{B}$ are independent of those in $\cup_{(c,i) \in \mathcal{C} \times [N_{\mathrm{e}}]} \mathcal{H}_{(c,i)}$, and that $\mathcal{G}_k$ only depends on samples in $\cup_{(c,i) \in \mathcal{C} \times [N_{\mathrm{e}}]} \mathcal{H}_{(c,i)}$, the regret bound of the stationary bandit. We also apply the fact that $R_{\mathcal{B}}(T)$ is increasing with $T$. Then, we can plug (23) and (22)

into (4) and obtain:

$$R_T$$

$$\leq \sum_{k=1}^{N_T+1} \bar{\Delta} \left(\nu_k - \nu_{k-1}\right) \left(|\mathcal{C}|N_e k \delta_F + (k-1)\left(2T^{-1} + \delta_D\right)\right)$$

$$+ \sum_{k=1}^{N_T+1} \left(\bar{\Delta}\ell_{k-1} + \bar{\Delta}\left[\bar{\alpha}\left(\nu_k - \nu_{k-1}\right) + N_e\right] + R_{\mathcal{B}}\left(\nu_k - \nu_{k-1}\right)\right)$$

$$\leq \sum_{k=1}^{N_T+1} \bar{\Delta}\left(\nu_k - \nu_{k-1}\right)\left(|\mathcal{C}|N_e\left(N_T+1\right)\delta_F + N_T\left(2T^{-1} + \delta_D\right)\right)$$

$$+ \sum_{k=1}^{N_T+1}\left(\bar{\Delta}\ell_{k-1} + \bar{\Delta}\left[\bar{\alpha}\left(\nu_k - \nu_{k-1}\right) + N_e\right] + R_{\mathcal{B}}\left(\nu_k - \nu_{k-1}\right)\right)$$

$$= \bar{\Delta}T|\mathcal{C}|N_e\left(N_T+1\right)\delta_F + 2\bar{\Delta}N_T + \bar{\Delta}TN_T\delta_D + \bar{\Delta}\sum_{k=1}^{N_T}\ell_k + \bar{\Delta}\left[\bar{\alpha}T + \left(N_T+1\right)N_e\right]$$

$$+ \sum_{k=1}^{N_T+1} R_{\mathcal{B}}\left(\nu_k - \nu_{k-1}\right)$$

$$\overset{(a)}{\leq} \bar{\Delta}T|\mathcal{C}|N_e\left(N_T+1\right)\delta_F + 2\bar{\Delta}N_T + \bar{\Delta}TN_T\delta_D + \bar{\Delta}\sum_{k=1}^{N_T}\ell_k + \bar{\Delta}\left[\bar{\alpha}T + \left(N_T+1\right)N_e\right]$$

$$+ \left(N_T+1\right)R_{\mathcal{B}}\left(\frac{T}{N_T+1}\right). \tag{24}$$

In step $(a)$, we apply Jensen's inequality to the concave function $R_{\mathcal{B}}$. This concludes the proof of Theorem 4.8.

### B.6. Proof of Corollary 4.9

In PS-PBs, $N_e = d$, $p \geq 1/2$, and $q = r = 0$. Thus, $R_T = \tilde{\mathcal{O}}(\sqrt{dN_TT} + d^p\sqrt{N_TT}) = \tilde{\mathcal{O}}(d^p\gamma_T^q(\log|\Pi|)^r\sqrt{N_TT})$.

In PS-KBs, $q \geq 1/2$, $p \geq 0$ and $r = 0$. We can upper bound $N_e$ using the fact that $|\mathcal{V}_T| \leq \lceil\sqrt{d}R/2\delta_T\rceil^d$. Thus, $N_e \leq \lceil C\gamma_T^{2q/d}\rceil^d$ with $\delta_T = \frac{Rd^{1/2-2p/d}}{2(C\gamma_T^{2q})^{1/d}}$ and $R_T = \tilde{\mathcal{O}}((d^{2p}\gamma_T^{2q}N_TT)^{1/2} + d^p\gamma_T^q\sqrt{N_TT}) = \tilde{\mathcal{O}}(d^p\gamma_T^q(\log|\Pi|)^r\sqrt{N_TT})$.

We emphasize that when the number of action is smaller than the covering number, i.e., $|\mathcal{A}| < \lceil C\gamma_T^{2q/d}\rceil^d \leq \gamma_T$, then we can set $\mathcal{A}_e$ to be the entire action set $\mathcal{A}$. In this case, $N_e < \gamma_T$, guaranteeing order-optimal regret.

In PS-CBs, $N_e \leq |\mathcal{A}|$, $r \geq 1/2$, $p = q = 0$, and $|\Pi| = |\mathcal{A}|^{|\mathcal{C}|}$. Thus, $R_T = \tilde{\mathcal{O}}((|\mathcal{A}|\log|\Pi|)^r\sqrt{N_TT} + \sqrt{|\mathcal{C}||\mathcal{A}|N_TT}) = \tilde{\mathcal{O}}(d^p\gamma_T^q(|\mathcal{A}|\log|\Pi|)^r\sqrt{N_TT})$.

