# OpenReview forum: "DAL: A Practical Prior-Free Black-Box Framework for Piecewise Stationary Bandits"
_ICML.cc/2026/Conference — ICML 2026 regular_

### Official Review · Reviewer_Sib7 · 2026-02-22

**Soundness:** 3
**Presentation:** 1
**Significance:** 3
**Originality:** 2
**Overall Recommendation:** 4
**Confidence:** 4

**Summary:**

This paper studies piecewise stationary bandits and proposes Detection Augmented Learning (DAL), a practical, prior-free black-box framework. DAL modularly couples stationary bandit algorithm with a change detector (e.g., GLR/GSR) and a forced exploration scheme over a carefully constructed action covering set . Compared with existing order-optimal black-box methods (e.g., MASTER), DAL offers several advantages: it is easy to implement, requires no hyperparameter tuning, and can be directly applied across various bandit settings. The authors validate the approach on both synthetic and real-world datasets, showing lower dynamic regret than competing algorithms, and provide theoretical guarantees that match or improve best known regret bounds.

**Compliance With Llm Reviewing Policy:**

Affirmed.

**Final Justification:**

I appreciate the authors' response to my concerns. I have increased my score to 4.

**Key Questions For Authors:**

1. Could the authors discuss why practical frameworks have been lacking and elaborate on the specific challenges involved in designing the proposed algorithm? I believe addressing this would help clarify the core contributions of the paper.
2. The framework appears less practical for the contextual setting, since the covering set must span the entire action space. How does DAL scale in real-world scenarios (e.g., recommendation systems) with massive action spaces? Is there any potential to construct a smaller covering set, or is the framework inherently limited in this regard?
3. Could the authors clarify whether the algorithm remains valid when the total time horizon T is unknown in advance?

**Strengths And Weaknesses:**

**Strengths:**

1. The paper aims to provide a practical framework for non-stationary bandits. Existing frameworks such as MASTER are highly complex and difficult to deploy, so this direction is valuable for the community.
2. The paper includes extensive experiments demonstrating that the proposed algorithm achieves lower regret. The code is available, and after running it, I found that the overall conclusion that the method achieves competitive or lower regret is valid.

---

**Weaknesses:**

1. The current manuscript could benefit from substantial refinement to improve overall readability. For instance, the paper uses too many abbreviations. While this may not hinder readers who are very familiar with the non-stationary bandits literature, it is unnecessary and increases the cognitive burden for readers. In addition, the organization makes it difficult to identify the true contributions. The introduction spends substantial space on related work that is not closely connected to the paper, while the core contributions and challenges are not clearly articulated, such as why practical frameworks are lacking, what the real design and analysis challenges are, and what exactly the methodological and theoretical contributions of the paper are.
2. The theoretical analysis relies on Piecewise Stationary Bandits under Assumption 4.6, rather than addressing general non-stationary bandits.
3. Partially due to the presentation issues mentioned above, the precise technical novelty of the paper appears somewhat incremental. Because the framework leverages existing change detection tests (e.g., GLR/GSR) and focuses on a somewhat simplified version of piecewise stationary bandits, it is difficult to isolate the new technical developments. The authors could more explicitly highlight what novel analytical tools or algorithmic mechanisms they introduce beyond the modular combination of existing techniques.

---

Minor comment: The provided code records runtime, but it is unclear why runtime is not reported in the paper.

---

> ### Author Rebuttal · Authors · 2026-03-27
>
> We sincerely thank Reviewer Sib7 for their thoughtful feedback, the recognition of DAL's practical value, and the verification of our experimental results. Below we address all points.
>
> **W1 (Readability):** We appreciate the suggestion. We resorted to abbreviations for plot/table readability, consistency with the literature, and space constraints, but will significantly reduce their use in a revision. Regarding organization, the final two paragraphs of the introduction articulate the literature gaps and our contributions; we expand further in Q1 below.
>
> **W2 (Assumption 4.6 and general NS):** The lack of drift guarantees is intentional, as our focus is PS settings, with drifting experiments demonstrating empirical robustness. Regarding Assumption 4.6, we admit our analysis relies on it. However, none of our experiments enforce it, yet DAL dominates. We refer the reviewer to our response to Reviewer jdRw (W3), where we identify a potential issue in MASTER's proof and argue that a condition like Assumption 4.6 may be fundamental to detection-restart approaches that employ any detector.
>
> **W3 (Technical novelty):** We respectfully clarify that our contribution is not tied to the GLR/GSR tests. There are two main novelties. First, DAL is a black-box method that turns any order-optimal stationary algorithm into an order-optimal piecewise stationary one by augmenting it with any detector satisfying Property 4.1. We present both GLR and GSR to demonstrate the existence of tests with Property 4.1 and the flexibility in detector choice. Second, compared to all existing prior-free methods, we detect actual changes in the reward model instead of violations of stationary regret bounds, which also performs much better empirically. This "detect-the-model" paradigm is fundamentally different from all prior work and enables DAL's practical effectiveness.
>
> The simple algorithmic structure reduces computational complexity, but does not make the theoretical analysis trivial: the contextual bandit generalization requires careful proof construction; the covering set must detect all changes while ensuring optimal regret growth. Through this, DAL matches or improves state-of-the-art bounds (Table 1) and provides a framework readily generalizable to settings such as MDPs. We also stress that the robust empirical performance across various real-world datasets is itself a significant contribution to the bandit community.
>
> **Q1 (Why practical frameworks are lacking):** The majority of approaches in the literature are prior-based, and since non-stationarity cannot be known in advance, these methods are impossible to employ in practice. The only prior-free alternatives detect non-stationarity by checking violations of stationary regret bounds. As shown in Gerogiannis et al. 2025, this leads to infeasibility, as MASTER requires a horizon of at least 143 billion for two-armed bandits. DAL addresses this gap by detecting changes in the reward model itself. Key design challenges are: (i) identifying what signal to detect in a prior-free setting, (ii) determining detector properties (Property 4.1) for order-optimality, including the balance of pre- and post-change sample complexity, and (iii) selecting which environment subset to monitor, particularly in infinite-action settings where the covering set construction (Propositions 4.2, 4.3) must balance exploration cost against detection guarantees.
>
> **Q2 (Contextual scaling):** We refer the reviewer to Remark 4.4 and emphasize that the covering set spans the entire action set only when there is no underlying reward structure, which is reasonable since $\mathcal{A}_{e} = \mathcal{A}$ in a multi-armed bandit (MAB), a special case of contextual bandits. Hence, in contextual settings with structure, we can use the propositions of our work. DAL is not constrained by design but by the setting's inherent complexity. The forced exploration matches the information-theoretic lower bound: in contextual bandits without structure, the base algorithm's complexity already depends on the total number of actions. Domain knowledge suggesting arm similarity can reduce $\mathcal{A}_e$ in practice. See also our response to Reviewer dm3c (W2).
>
> **Q3 (Unknown $T$):** Yes, DAL extends readily to unknown horizons via the doubling trick, also used in MASTER. We omitted this since compared methods are prior-based and require both $T$ and non-stationarity parameters, so DAL's advantage in not needing non-stationarity knowledge was the more pressing point. Given additional space, we will include this.
>
> **Minor (Runtime):** As noted in Appendix A.3, each experiment took under one hour, and our approach was notably efficient. We omitted runtime due to space but will include it given additional space. We thank the reviewer for confirming our results.
>
> ---
>
> We hope to have addressed all questions. We welcome further feedback and hope the reviewer will kindly consider our clarifications and contributions.

---

> > ### Author Rebuttal · Reviewer_Sib7 · 2026-03-31
> >
> > I thank the authors for their response, which has addressed most of my concerns. I hope the authors will incorporate these discussions into the revised manuscript. I will increase my score.

---

> > > ### Author Response · Authors · 2026-03-31
> > >
> > > Dear Reviewer Sib7,
> > >
> > > Thank you very much for your positive update and for recognizing the contributions of our work.
> > >
> > > Thank you for your feedback, we will incorporate all these discussions in our revised manuscript.
> > >
> > > We truly appreciate your constructive feedback and support.

---

### Official Review · Reviewer_dm3c · 2026-03-05

**Soundness:** 3
**Presentation:** 3
**Significance:** 3
**Originality:** 3
**Overall Recommendation:** 4
**Confidence:** 3

**Summary:**

This paper introduces Detection Augmented Learning (DAL), a modular black-box framework designed for piecewise stationary bandits where the timing and nature of environment changes are unknown. The framework augments any order-optimal stationary bandit algorithm with a change detector and a forced exploration scheme to handle non-stationarity. Unlike existing methods that monitor regret violations, DAL detects shifts in mean rewards through a carefully selected finite covering set of actions. The authors provided theoretical regret bounds for DAL, as well as empirically evaluated it across various datasets to show its effectiveness.

**Compliance With Llm Reviewing Policy:**

Affirmed.

**Key Questions For Authors:**

See weaknesses.

**Limitations:**

Yes.

**Strengths And Weaknesses:**

Strength:
- The introduced DAL framework here is very flexible as it can be paired with any stationary bandit algorithm, making it applicable to a wide array of bandit variants.
- The authors conducted very extensive numerical evaluations here that considered a number of non-stationary bandits settings, including synthetic piecewise stationary & drifting regimes and a suite of real-world benchmarks.

Weakness:
- The theoretical optimality of DAL still relies on Assumption 4.6, which requires a minimum spacing between change-points.
- The framework requires forced exploration on a covering set $\mathcal{A}_e$, and in complex action spaces where the required size of $\mathcal{A}_e$ is large, the computational overhead can be large.
- The authors suggested that small drift can be “absorbed into noise", but there’s no formal regret guarantee for drift. Is there a formal way to characterize the "critical" drift rate at which a restart-based method like DAL becomes mathematically superior to a sliding window type of method?

---

> ### Author Rebuttal · Authors · 2026-03-27
>
> We sincerely thank Reviewer dm3c for their thoughtful and constructive feedback. We appreciate the recognition of DAL's flexibility and the breadth of the empirical evaluation. Below we address the main concerns.
>
> **W1 (Assumption 4.6):** We agree that Assumption 4.6 is a limitation from a theoretical prior-free perspective. In our analysis, this assumption is needed to guarantee that the detector has enough pre- and post-change samples to reliably trigger a restart, i.e., it ensures that the change detector can observe sufficiently many samples for Property 4.1 to apply. It is possible to generalize the analysis to cases where Assumption 4.6 does not hold, but it will require regret analysis of stationary bandit algorithms in nonstationary environments, as change detectors might fail to detect changes. This regret analysis, however, is highly non-trivial and out of the scope for our work.
>
> We would like to emphasize two important points: (i) we argue that this requirement is fundamental to any explicit change-detection method, i.e., reliably identifying distributional shifts requires sufficient samples around each change. Any method that aims to explicitly identify all changes in the data-generating process faces this constraint; (ii) as discussed in Remark 4.7, Assumption 4.6 is *not enforced* in any of our experiments, including the real-world ones, yet DAL still dominates all baselines.
>
> We believe DAL is substantially more practical than the only prior-free black-box alternative, MASTER, while also being conceptually different: DAL detects mean-shifts in the reward process itself, rather than monitoring violations of stationary regret guarantees. As shown in Gerogiannis et al. 2025, MASTER theoretically requires a horizon of at least 143 billion for the simplest two-armed bandit setting, making it infeasible in practice. We also refer the reviewer to our response to Reviewer jdRw W3, where we identify a potential issue in MASTER's proof of order-optimality, further contextualizing the significance of our contribution.
>
> **W2 (Exploration overhead):** We would like to emphasize that when $|\mathcal{A}_ {e}|$ is large, the non-stationary bandit problem itself is already difficult. To see this, consider an algorithm that restarts perfectly in piecewise stationary parametric bandits where $|\mathcal{A}_ {e}|$ is large. The regret of this perfect-restart algorithm is $\tilde{\mathcal{O}}(d\sqrt{N_ {T}T})$ where $d \geq |\mathcal{A}_ {e}|$. Since the regret incurred by forced exploration is $\tilde{\mathcal{O}}(\sqrt{|\mathcal{A}_ {e}|N_ {T}T})$, it does not contribute significant overhead. A similar argument extends to piecewise stationary kernelized and contextual bandits. We also stress that forced exploration does not impose large computational complexity, as action selection is trivial and the change detector monitors only one action per time-step.
>
> More broadly, DAL is driven by the structural complexity of the reward class, not by the raw cardinality of the action space. In parametric bandits, $|\mathcal{A}_e| \le d$ (Proposition 4.2). In kernelized bandits, $|\mathcal{A}_e|$ is controlled by the cover construction (Proposition 4.3/Corollary 4.9). The least favorable case is the contextual setting without structure, where $\mathcal{A}_e = \mathcal{A}$ may be needed, but this reflects the inherent difficulty of prior-free detection when an arbitrary change may affect any action. Further details appear in Appendix A.1.
>
> **W3 (Formal drift characterization):** We do not provide a formal regret comparison between restart-based and sliding-window methods under drift. In Section 4.3, our goal was more modest: to provide intuition and experiments for why DAL performs well under drift, even though our formal guarantees are for piecewise stationary environments. That said, in the linear setting of Section 4.3, a natural heuristic emerges. The drift-induced perturbation $\langle \zeta_{t+1}, a \rangle$ has variance at most $\delta^2 L^2/(d+2)$, so a critical scale is when this matches the noise variance $\sigma^2$:
>
> $$\delta_{\mathrm{crit}} \asymp \frac{\sqrt{d+2}}{L} \sigma.$$
>
> When $\delta \ll \delta_{\mathrm{crit}}$, the one-step effect of the drift is statistically comparable to noise, which helps explain why a restart-based method such as DAL can still behave well in practice. When $\delta$ grows, the cumulative bias increasingly favors adaptive windowing/discounting, consistent with Figure 3. We emphasize, however, that this should be interpreted as a heuristic threshold rather than a formal superiority guarantee. Establishing a sharp regret-level crossover between restart-based and sliding-window methods in drifting environments is an important open direction beyond our current scope.
>
> ---
>
> We hope to have addressed all concerns. We welcome further feedback and hope the reviewer will kindly consider our clarifications and contributions. We once again thank the reviewer for the positive support.

---

> > ### Author Rebuttal · Reviewer_dm3c · 2026-04-03
> >
> > Thank you for your detailed response. I will keep my evaluation of weak acceptance.

---

> > > ### Author Response · Authors · 2026-04-03
> > >
> > > Dear Reviewer dm3c,
> > >
> > > Thank you very much for your positive update and for recognizing the contributions of our work.
> > >
> > > We truly appreciate your constructive feedback and support.

---

### Official Review · Reviewer_HhGQ · 2026-03-12

**Soundness:** 3
**Presentation:** 2
**Significance:** 3
**Originality:** 3
**Overall Recommendation:** 5
**Confidence:** 3

**Summary:**

This paper introduces DAL, a prior-free black-box framework for piecewise stationary bandits that augments any order-optimal stationary bandit algorithm with change detection and forced exploration. It addresses non-stationarity without prior knowledge, outperforming baselines in synthetic and real-world settings.

**Compliance With Llm Reviewing Policy:**

Affirmed.

**Key Questions For Authors:**

see weaknesses

**Limitations:**

Yes they discuss sufficient limitations and future work in the conclusion

**Strengths And Weaknesses:**

Strengths:
+ Paper addresses a practical and important problem and covers most of the important references as far as I can understand
+ The introduction of the change point detector regardless of the input bandit algorithm can be quite useful for many practical scenarios and can be a bridging path between many theoretical Bandit algorithms and real-world applications
+ Empirical results validate their claims

Weaknesses:
- How costly can be computing Proposition 4.2?
- The paper is a quite dense, and makes it a little hard to understand

---

> ### Author Rebuttal · Authors · 2026-03-27
>
> We sincerely thank Reviewer HhGQ for their thoughtful and encouraging feedback. We greatly appreciate the reviewer's recognition of the practical relevance of the problem, the usefulness of a detector-augmented black-box design, and the strength of the empirical validation. We are also grateful for the constructive comments regarding the computational cost of Proposition 4.2 and the density of the presentation. Below, we address the main points raised.
>
> **W1 (Cost of Proposition 4.2):** The computation in Proposition 4.2 is a *one-time preprocessing step* carried out before the online execution of DAL, and is therefore not incurred at every round. In the parametric setting, the construction of the covering set is straightforward: one greedily scans the candidate actions and retains an action whenever it is linearly independent of those already selected, until the covering set spans the action space. Thus, the practical cost scales with the number of candidate actions inspected together with the corresponding linear-independence checks, rather than with the horizon $T$. Importantly, this overhead depends on the structural diversity of the action space and is typically negligible relative to the cost of the full learning procedure, especially when the horizon is reasonably large.
>
> In particular, the role of Proposition 4.2 is mainly structural: once $\mathcal{A}_e$ is chosen as a maximal linearly independent subset, the parameter $\theta_t$ is uniquely determined by the inner products $\lbrace\langle \theta_t, a \rangle : a \in \mathcal{A}_e\rbrace$. Therefore, any change in $\theta_t$ must induce a detectable change on at least one action in $\mathcal{A}_e$.
>
> Some of these implementation details were moved to the appendix due to limited space in the main paper. In particular, Appendix A.1 discusses how $\mathcal{A}_e$ is constructed in practice in finite-action settings, and the appendix also provides the proof details underlying Proposition 4.2. We agree that this point could be made more explicit in the main text, and we will clarify more clearly that this construction is performed only once as preprocessing and does not constitute a per-round computational burden.
>
> **W2 (Density of presentation):** We agree that the paper is dense, in part because we aim to present a unified framework together with guarantees and experiments across several different bandit settings. Our goal was to balance generality, theoretical completeness, and space limitations. Due to limited space, several implementation and proof details were moved to the appendix. This includes further discussion of the construction of $\mathcal{A}_e$, practical considerations in finite action spaces, and some of the proof details for the main propositions and theorem.
>
> We would be very grateful for any more specific suggestions on which sections or points were hardest to follow. This would help us improve the exposition most effectively in a revised version. More broadly, we will aim to improve the high-level guidance and signposting in the paper so that the intuition behind DAL and the role of the covering set $\mathcal{A}_e$ are easier to follow.
>
> ---
>
> We are grateful for the reviewer's positive assessment and constructive suggestions, and we thank them again for their support.

---

> > ### Author Rebuttal · Reviewer_HhGQ · 2026-04-03
> >
> > I thank the authors for addresing my concern.

---

> > > ### Author Response · Authors · 2026-04-03
> > >
> > > Dear Reviewer HhGQ,
> > >
> > > Thank you very much for your positive score and for recognizing the contributions of our work.
> > >
> > > We truly appreciate your constructive feedback and support.

---

### Official Review · Reviewer_jdRw · 2026-03-25

**Soundness:** 3
**Presentation:** 2
**Significance:** 3
**Originality:** 3
**Overall Recommendation:** 4
**Confidence:** 3

**Summary:**

The authors consider piecewise-stationary (PS) bandit framework without the prior knowledge of underlying non-stationarity and propose a DAL (black-box) method which achieves order-optimal regret under certain assumptions. The claims made in the paper are supported with strong empirical performance validated using synthetic and real-world experiments.

**Compliance With Llm Reviewing Policy:**

Affirmed.

**Final Justification:**

Fully resolved. My concerns have been adequately addressed. I have updated the scores. My final recommendation is weak accept.

**Key Questions For Authors:**

NA

**Limitations:**

The paper does not adequately discuss the  limitations of method.

**Strengths And Weaknesses:**

Strengths:
1.	The proposed DAL method is free from prior knowledge of non-stationarity, with theoretical guarantees and strong empirical performance.
2.	The framework is also versatile, covering a wide range of bandit settings, increasing its applicability.

Weakness: Some limitations are:
1.	The practical applicability of the proposed method/algorithm seem very challenging.
2.	The paper shows empirical performance for drifting case. However, no theoretical guarantee for this case is provided.
3.	The theoretical results rely on assumptions such as sufficient separation between change-points and the detectability of reward shifts, which may not always hold in practical settings, especially in environments with frequent or subtle changes.

---

> ### Author Rebuttal · Authors · 2026-03-27
>
> We sincerely thank Reviewer jdRw for their feedback and the recognition that DAL is prior-free, theoretically grounded, and empirically strong. Below we address all concerns.
>
> **W1 (Practical applicability):** We would like to respectfully disagree with this characterization. As our title already suggests, a central goal of our work is not only to advance the theory of non-stationary bandits, but also to investigate and improve their practical applicability in real-world settings. This is precisely why we place strong emphasis on extensive empirical validation.
>
> Concretely, we conduct 20 synthetic experiments across diverse non-stationary environments, which, to the best of our knowledge, is substantially broader than what all prior work has attempted, and 9 real-world experiments on standard datasets drawn from well-known engineered applications. DAL consistently outperforms all prior-free and prior-based baselines. We believe this level of empirical validation is precisely what supports the practical applicability claim.
>
> Moreover, the dataset of Gerogiannis & Torrellas (2023) corresponds to an actual real-world application where the problem is naturally formulated as a bandit problem. DAL improves upon their benchmark performance. In Appendix A.2 we visualize this data, arguing that the environment appears much more piecewise-stationary than drifting, further motivating our method. Our framework is intentionally modular and plug-and-play: it augments any stationary bandit algorithm with a detector and a forced exploration scheme, without requiring prior knowledge about the non-stationarity. Section 3.2 describes its practical instantiation across all settings.
>
> If the reviewer found particular components especially challenging from a practical perspective, we would be grateful for more specific feedback, as this would help us clarify the exposition more effectively.
>
> **W2 (No drift theory):** We agree, and this is intentional and consistent with the scope of the paper. As stated in the title, introduction, and main text, our focus is the PS setting. The drifting experiments highlight an empirical finding relevant to practitioners, not a theorem-level contribution. We explicitly list drift guarantees as an open direction in Section 5. We refer the reviewer to our response to Reviewer dm3c (W3), where we provide a heuristic characterization of when DAL excels under drift.
>
> **W3 (Assumptions may not hold):** We agree Assumption 4.6 restricts the theory, but it is explicitly discussed in Remark 4.7 and Section 5. None of our experiments enforce it, yet DAL dominates, including settings with frequent, arbitrarily placed change-points ($\xi = 0.4$). The reason this assumption appears is fundamental to any explicit change-detection method: reliably identifying distributional shifts requires sufficiently many pre- and post-change samples.
>
> At the same time, the only existing prior-free black-box alternative is MASTER. As shown in Gerogiannis et al. 2025, for MASTER's detection mechanism to work even in the simplest two-armed bandit case, the required horizon would need to be at least 143 billion, something that is far beyond any realistic practical regime.
>
> Moreover, during the rebuttal period, we identified what appears to be a substantive issue in MASTER's proof of order-optimality. In the proof of Lemma 17, the authors state that **term3** counts trials up to the first success with probability $\alpha_n/\alpha_m$. However:
>
> $$\textbf{term3} = \sum_{t=t_n}^{t_n+2^n-1} \textbf{1} \left\lbrace W_t \cap Y_t \cap \bigcap_{\tau=t_n}^{t} \bar{W}_{\tau} \cup \bar{Y} _{\tau} \cup \bar{Z} _{\tau} \right\rbrace$$
>
> which counts candidate points where either no order-$m$ algorithm is scheduled *or* $W_\tau$ fails, which is strictly larger than claimed. If correct, MASTER's order-optimality may not follow as stated, potentially making DAL the only prior-free black-box framework with a valid order-optimal guarantee. We believe this further strengthens the significance of our contribution.
>
> **Limitations:** We respectfully note that Section 5 explicitly identifies two main limitations: (i) the lack of drift regret guarantees, and (ii) reliance on a change-point separation condition, which limits the extent to which DAL achieves fully prior-free theoretical optimality. Some supporting details were moved to the appendix due to space, but the main limitations are clearly stated in the paper.
>
> ---
>
> We hope to have addressed all of the reviewer’s questions to the best of our ability. Nevertheless, we would be more than happy to receive further feedback if there are additional concerns. We hope the reviewer will kindly consider our clarifications and the contributions of our work.

---

> > ### Author Rebuttal · Reviewer_jdRw · 2026-04-03
> >
> > I thank the authors for clearly clarifying my concerns. My overall recommendation is weak accept.

---

> > > ### Author Response · Authors · 2026-04-03
> > >
> > > Dear Reviewer jdRw,
> > >
> > > Thank you very much for your positive update and for recognizing our work. We really appreciate it!
> > >
> > > We wanted to check if there are any remaining concerns that we may not have fully addressed. We would be happy to clarify anything further and update our response while the discussion period is still ongoing.
> > >
> > > We also noticed that the score has not been updated, so we just wanted to kindly ask if it reflects your current assessment or if there is anything else we can improve or address.
> > >
> > > Thank you again for your time and feedback.

---

### Decision · Program_Chairs · 2026-04-30

**Decision:**

Accept (regular)

**Comment:**

This paper proposes a black-box framework for piecewise stationary bandits that combines change detection with a stationary bandit algorithm as a plug-in base learner. A major strength of the paper is its strong empirical performance, supported by extensive experiments across a diverse set of datasets and environments. The main concern raised by the reviewers is that the theoretical guarantees rely on assumptions on change-point separation, and the analysis does not directly extend to the drifting setting. The authors are encouraged to clarify these limitations more explicitly in the final version.